# Reducing Noise in GAN Training with Variance Reduced Extragradient

**Tatjana Chavdarova**[*]
Mila, Université de Montréal
Idiap, École Polytechnique Fédérale de Lausanne

**Gauthier Gidel**[*]
Mila, Université de Montréal
Element AI

**François Fleuret**
Idiap, École Polytechnique Fédérale de Lausanne

**Simon Lacoste-Julien**[†]
Mila, Université de Montréal

## Abstract

We study the effect of the stochastic gradient noise on the training of generative adversarial networks (GANs) and show that it can prevent the convergence of standard game optimization methods, while the batch version converges. We address this issue with a novel stochastic variance-reduced extragradient (SVRE) optimization algorithm, which for a large class of games improves upon the previous convergence rates proposed in the literature. We observe empirically that SVRE performs similarly to a batch method on MNIST while being computationally cheaper, and that SVRE yields more stable GAN training on standard datasets.

## 1 Introduction

Many empirical risk minimization algorithms rely on gradient-based optimization methods. These iterative methods handle large-scale training datasets by computing gradient estimates on a subset of it, a *mini-batch*, instead of using all the samples at each step, the *full batch*, resulting in a method called *stochastic gradient descent* (SGD, Robbins and Monro (1951); Bottou (2010)).

SGD methods are known to efficiently minimize *single* objective loss functions, such as cross-entropy for classification or squared loss for regression. Some algorithms go beyond such training objective and define multiple agents with different or competing objectives. The associated optimization paradigm requires a multi-objective joint minimization. An example of such a class of algorithms are the generative adversarial networks (GANs, Goodfellow et al., 2014), which aim at finding a Nash equilibrium of a two-player *minimax* game, where the players are deep neural networks (DNNs).

As of their success on supervised tasks, SGD based algorithms have been adopted for GAN training as well. Recently, Gidel et al. (2019a) proposed to use an optimization technique coming from the variational inequality literature called *extragradient* (Korpelevich, 1976) with provable convergence guarantees to optimize games (see § 2). However, convergence failures, poor performance (sometimes referred to as "mode collapse"), or hyperparameter susceptibility are more commonly reported compared to classical supervised DNN optimization.

We question naive adoption of such methods for game optimization so as to address the reported training instabilities. We argue that as of the two player setting, noise impedes drastically more the training compared to single objective one. More precisely, we point out that the noise due to the stochasticity may break the convergence of the extragradient method, by considering a simplistic stochastic bilinear game for which it provably does *not* converge.

---

[*]equal contribution
[†]Canada CIFAR AI Chair

| Method | Complexity | $\mu$-adaptivity |
|--------|-----------|------------------|
| SVRG | $\ln(\frac{1}{\epsilon})\times(n+\frac{\bar{L}^2}{\mu^2})$ | no |
| Acc. SVRG | $\ln(\frac{1}{\epsilon})\times(n+\sqrt{n}\frac{\bar{L}}{\mu})$ | no |
| **SVRE** §3.2 | $\ln(\frac{1}{\epsilon})\times(n+\frac{\bar{\ell}}{\mu})$ | if $\bar{\ell}=O(\bar{L})$ |

Table 1: Comparison of variance reduced methods for games for a $\mu$-strongly monotone operator with $L_i$-Lipschitz stochastic operators. Our result makes the assumption that the operators are $\ell_i$-cocoercive. Note that $\ell_i \in [L_i, L_i^2/\mu]$, more details and a tighter rate are provided in §3.2. The SVRG variants are proposed by Palaniappan and Bach (2016). *$\mu$-adaptivity* indicates if the hyper-parameters that guarantee convergence (step size & epoch length) depend on the strong monotonicity parameter $\mu$: if not, the algorithm is adaptive to local strong monotonicity. Note that in some cases the constant $\ell$ may depend on $\mu$ but SVRE is adaptive to strong convexity when $\bar{\ell}$ remains close to $\bar{L}$ (see for instance Proposition 2).

**Algorithm 1** Pseudocode for SVRE.

1: **Input:** Stopping time $T$, learning rates $\eta_{\boldsymbol{\theta}}, \eta_{\boldsymbol{\varphi}}$, initial weights $\boldsymbol{\theta}_0, \boldsymbol{\varphi}_0$. $t = 0$
2: **while** $t \le T$ **do**
3: $\quad \boldsymbol{\varphi}^{\mathcal{S}} = \boldsymbol{\varphi}_t$ and $\boldsymbol{\mu}_{\boldsymbol{\varphi}}^{\mathcal{S}} = \frac{1}{n}\sum_{i=1}^{n}\nabla_{\boldsymbol{\varphi}}\mathcal{L}_i^D(\boldsymbol{\theta}^{\mathcal{S}}, \boldsymbol{\varphi}^{\mathcal{S}})$
4: $\quad \boldsymbol{\theta}^{\mathcal{S}} = \boldsymbol{\theta}_t$ and $\boldsymbol{\mu}_{\boldsymbol{\theta}}^{\mathcal{S}} = \frac{1}{n}\sum_{i=1}^{n}\nabla_{\boldsymbol{\theta}}\mathcal{L}_i^G(\boldsymbol{\theta}^{\mathcal{S}}, \boldsymbol{\varphi}^{\mathcal{S}})$
5: $\quad N \sim \text{Geom}\left(1/n\right)$ *(Sample epoch length)*
6: $\quad$ **for** $i = 0$ **to** $N-1$ **do** {*Beginning of the epoch*}
7: $\quad\quad$ **Sample** $i_{\boldsymbol{\theta}}, i_{\boldsymbol{\varphi}} \sim \pi_{\boldsymbol{\theta}}, \pi_{\boldsymbol{\varphi}}$, do **extrapolation:**
8: $\quad\quad \tilde{\boldsymbol{\varphi}}_t = \boldsymbol{\varphi}_t - \eta_{\boldsymbol{\varphi}}\boldsymbol{d}_{i_{\boldsymbol{\varphi}}}^D(\boldsymbol{\theta}_t, \boldsymbol{\varphi}_t, \boldsymbol{\theta}^{\mathcal{S}}, \boldsymbol{\varphi}^{\mathcal{S}}) \quad \triangleright (5)$
9: $\quad\quad \tilde{\boldsymbol{\theta}}_t = \boldsymbol{\theta}_t - \eta_{\boldsymbol{\theta}}\boldsymbol{d}_{i_{\boldsymbol{\theta}}}^G(\boldsymbol{\theta}_t, \boldsymbol{\varphi}_t, \boldsymbol{\theta}^{\mathcal{S}}, \boldsymbol{\varphi}^{\mathcal{S}}) \quad \triangleright (5)$
10: $\quad\quad$ **Sample** $i_{\boldsymbol{\theta}}, i_{\boldsymbol{\varphi}} \sim \pi_{\boldsymbol{\theta}}, \pi_{\boldsymbol{\varphi}}$ and do **update:**
11: $\quad\quad \boldsymbol{\varphi}_{t+1} = \boldsymbol{\varphi}_t - \eta_{\boldsymbol{\varphi}}\boldsymbol{d}_{i_{\boldsymbol{\varphi}}}^D(\tilde{\boldsymbol{\theta}}_t, \tilde{\boldsymbol{\varphi}}_t, \boldsymbol{\theta}^{\mathcal{S}}, \boldsymbol{\varphi}^{\mathcal{S}}) \quad \triangleright (5)$
12: $\quad\quad \boldsymbol{\theta}_{t+1} = \boldsymbol{\theta}_t - \eta_{\boldsymbol{\theta}}\boldsymbol{d}_{i_{\boldsymbol{\theta}}}^G(\tilde{\boldsymbol{\theta}}_t, \tilde{\boldsymbol{\varphi}}_t, \boldsymbol{\theta}^{\mathcal{S}}, \boldsymbol{\varphi}^{\mathcal{S}}) \quad \triangleright (5)$
13: $\quad\quad t \leftarrow t + 1$
14: **Output:** $\boldsymbol{\theta}_T, \boldsymbol{\varphi}_T$

The theoretical aspect we present in this paper is further supported empirically, since using larger mini-batch sizes for GAN training has been shown to considerably improve the quality of the samples produced by the resulting generative model: Brock et al. (2019) report a relative improvement of $46\%$ of the Inception Score metric (see § 4) on ImageNet if the batch size is increased 8–fold. This notable improvement raises the question if noise reduction optimization methods can be extended to game settings. In turn, this would allow for a principled training method with the practical benefit of omitting to empirically establish this multiplicative factor for the batch size.

In this paper, we investigate the interplay between noise and multi-objective problems in the context of GAN training. Our contributions can be summarized as follows: (i) we show in a motivating example how the noise can make stochastic extragradient fail (see § 2.2). (ii) we propose a new method "stochastic variance reduced extragradient" (SVRE) that combines variance reduction and extrapolation (see Alg. 1 and § 3.2) and show experimentally that it effectively reduces the noise. (iii) we prove the convergence of SVRE under local strong convexity assumptions, improving over the known rates of competitive methods for a large class of games (see § 3.2 for our convergence result and Table 1 for comparison with standard methods). (iv) we test SVRE empirically to train GANs on several standard datasets, and observe that it can improve SOTA deep models in the late stage of their optimization (see § 4).

## 2 GANs as a Game and Noise in Games

### 2.1 Game theory formulation of GANs

The models in a GAN are a generator $G$, that maps an embedding space to the signal space, and should eventually map a fixed noise distribution to the training data distribution, and a discriminator $D$ whose purpose is to allow the training of the generator by classifying genuine samples against generated ones. At each iteration of the algorithm, the discriminator $D$ is updated to improve its "real vs. generated" classification performance, and the generator $G$ to degrade it.

From a game theory point of view, GAN training is a differentiable two-player game where the generator $G_{\boldsymbol{\theta}}$ and the discriminator $D_{\boldsymbol{\varphi}}$ aim at minimizing their own cost function $\mathcal{L}^G$ and $\mathcal{L}^D$, resp.:

$$\boldsymbol{\theta}^* \in \arg\min_{\boldsymbol{\theta}\in\Theta} \mathcal{L}^G(\boldsymbol{\theta}, \boldsymbol{\varphi}^*) \qquad \text{and} \qquad \boldsymbol{\varphi}^* \in \arg\min_{\boldsymbol{\varphi}\in\Phi} \mathcal{L}^D(\boldsymbol{\theta}^*, \boldsymbol{\varphi}) \,. \tag{2P-G}$$

When $\mathcal{L}^D = -\mathcal{L}^G =: \mathcal{L}$ this game is called a *zero-sum game* and (2P-G) is a minimax problem:

$$\min_{\boldsymbol{\theta}\in\Theta}\max_{\boldsymbol{\varphi}\in\Phi} \mathcal{L}(\boldsymbol{\theta}, \boldsymbol{\varphi}) \tag{SP}$$

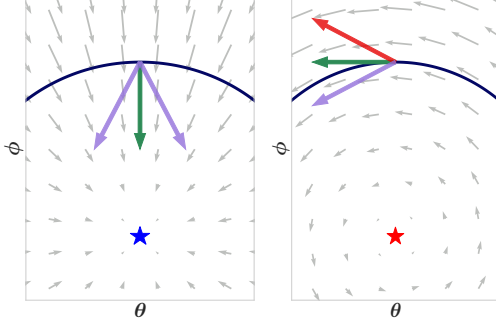

Figure 1: Illustration of the discrepancy between games and minimization on simple examples:

$$\textit{min:} \quad \min_{\theta,\phi\in\mathbb{R}} \theta^2 + \phi^2, \quad \textit{game:} \quad \min_{\theta\in\mathbb{R}} \max_{\phi\in\mathbb{R}} \theta \cdot \phi.$$

**Left: Minimization.** Up to a neighborhood, the noisy gradient always points to a direction that make the iterate closer to the minimum ($\star$).
**Right: Game.** The noisy gradient may point to a direction (red arrow) that push the iterate away from the Nash Equilibrium ($\star$).

The gradient method does not converge for some convex-concave examples (Mescheder et al., 2017; Gidel et al., 2019a). To address this, Korpelevich (1976) proposed to use the *extragradient* method[3] which performs a lookahead step in order to get signal from an *extrapolated* point:

$$\text{Extrapolation:} \begin{cases} \tilde{\boldsymbol{\theta}}_t = \boldsymbol{\theta}_t - \eta\nabla_{\boldsymbol{\theta}}\mathcal{L}^G(\boldsymbol{\theta}_t, \boldsymbol{\varphi}_t) \\ \tilde{\boldsymbol{\varphi}}_t = \boldsymbol{\varphi}_t - \eta\nabla_{\boldsymbol{\varphi}}\mathcal{L}^D(\boldsymbol{\theta}_t, \boldsymbol{\varphi}_t) \end{cases} \quad \text{Update:} \begin{cases} \boldsymbol{\theta}_{t+1} = \boldsymbol{\theta}_t - \eta\nabla_{\boldsymbol{\theta}}\mathcal{L}^G(\tilde{\boldsymbol{\theta}}_t, \tilde{\boldsymbol{\varphi}}_t) \\ \boldsymbol{\varphi}_{t+1} = \boldsymbol{\varphi}_t - \eta\nabla_{\boldsymbol{\varphi}}\mathcal{L}^D(\tilde{\boldsymbol{\theta}}_t, \tilde{\boldsymbol{\varphi}}_t) \end{cases} \quad \text{(EG)}$$

Note how $\boldsymbol{\theta}_t$ and $\boldsymbol{\varphi}_t$ are updated with a gradient from a different point, the *extrapolated* one. In the context of a zero-sum game, for any *convex-concave* function $\mathcal{L}$ and any closed convex sets $\Theta$ and $\Phi$, the extragradient method converges (Harker and Pang, 1990, Thm. 12.1.11).

## 2.2 Stochasticity Breaks Extragradient

As the (EG) converges for some examples for which gradient methods do not, it is reasonable to expect that so does its stochastic counterpart (at least to a neighborhood). However, the resulting noise in the gradient estimate may interact in a problematic way with the oscillations due to the *adversarial component* of the game[4]. We depict this phenomenon in Fig. 1, where we show the direction of the noisy gradient on single objective minimization example and contrast it with a multi-objective one.

We present a simplistic example where the extragradient method *converges linearly* (Gidel et al., 2019a, Corollary 1) using the full gradient but *diverges geometrically* when using stochastic estimates of it. Note that standard gradient methods, both batch and stochastic, diverge on this example.

In particular, we show that: (i) if we use standard stochastic estimates of the gradients of $\mathcal{L}$ with a simple finite sum formulation, then the iterates $\boldsymbol{\omega}_t := (\boldsymbol{\theta}_t, \boldsymbol{\varphi}_t)$ produced by the stochastic extragradient method (SEG) diverge geometrically, and on the other hand (ii) the full-batch extragradient method does converge to the Nash equilibrium $\boldsymbol{\omega}^*$ of this game (Harker and Pang, 1990, Thm. 12.1.11).

**Theorem 1** (Noise may induce divergence)**.** *For any $\epsilon \geq 0$ There exists a zero-sum $\frac{\epsilon}{2}$-strongly monotone stochastic game such that if $\boldsymbol{\omega}_0 \neq \boldsymbol{\omega}^*$, then for any step-size $\eta > \epsilon$, the iterates $(\boldsymbol{\omega}_t)$ computed by the stochastic extragradient method diverge geometrically, i.e., there exists $\rho > 0$, such that $\mathbb{E}[\|\boldsymbol{\omega}_t - \boldsymbol{\omega}^*\|^2] > \|\boldsymbol{\omega}_0 - \boldsymbol{\omega}^*\|^2(1 + \rho)^t$.*

*Proof sketch.* All detailed proofs can be found in § C of the appendix. We consider the following stochastic optimization problem (with $d = n$):

$$\frac{1}{n}\sum_{i=1}^{n} \frac{\epsilon}{2}\theta_i^2 + \boldsymbol{\theta}^\top \boldsymbol{A}_i\boldsymbol{\varphi} - \frac{\epsilon}{2}\varphi_i^2 \quad \text{where} \quad [\boldsymbol{A}_i]_{kl} = 1 \text{ if } k = l = i \text{ and } 0 \text{ otherwise.} \quad (1)$$

Note that this problem is a simple dot product between $\boldsymbol{\theta}$ and $\boldsymbol{\varphi}$ with an $(\epsilon/n)$-$\ell_2$ norm penalization, thus we can compute the batch gradient and notice that the Nash equilibrium of this problem is $(\boldsymbol{\theta}^*, \boldsymbol{\varphi}^*) = (\boldsymbol{0}, \boldsymbol{0})$. However, as we shall see, this simple problem *breaks* with standard stochastic optimization methods.

Sampling a mini-batch without replacement $I \subset \{1, \ldots, n\}$, we denote $\boldsymbol{A}_I := \sum_{i \in I} \boldsymbol{A}_i$. The extragradient update rule can be written as:

$$\begin{cases} \boldsymbol{\theta}_{t+1} = (1 - \eta \boldsymbol{A}_I \epsilon) \boldsymbol{\theta}_t - \eta \boldsymbol{A}_I ((1 - \eta \boldsymbol{A}_J \epsilon) \boldsymbol{\varphi}_t + \eta \boldsymbol{A}_J \boldsymbol{\theta}_t) \\ \boldsymbol{\varphi}_{t+1} = (1 - \eta \boldsymbol{A}_I \epsilon) \boldsymbol{\varphi}_t + \eta \boldsymbol{A}_I ((1 - \eta \boldsymbol{A}_J \epsilon) \boldsymbol{\theta}_t - \eta \boldsymbol{A}_J \boldsymbol{\varphi}_t) \,, \end{cases} \quad (2)$$

where $I$ and $J$ are the mini-batches sampled for the update and the extrapolation step, respectively. Let us write $N_t := \|\boldsymbol{\theta}_t\|^2 + \|\boldsymbol{\varphi}_t\|^2$. Noticing that $[\boldsymbol{A}_I \boldsymbol{\theta}]_i = [\boldsymbol{\theta}]_i$ if $i \in I$ and $0$ otherwise, we have,

$$\mathbb{E}[N_{t+1}] = \left( 1 - \frac{|I|}{n} (2\eta\epsilon - \eta^2(1 + \epsilon^2)) - \frac{|I|^2}{n^2} (2\eta^2 - \eta^4(1 + \epsilon^2)) \right) \mathbb{E}[N_t] \,. \quad (3)$$

Consequently, if the mini-batch size is smaller than half of the dataset size, i.e. $2|I| \leq n$, we have that $\forall \eta > \epsilon \,, \exists \rho > 0 \,, s.t. \,, \mathbb{E}[N_t] > N_0(1 + \rho)^t$. For the theorem statement, we set $n = 2$ and $|I| = 1$.

This result may seem contradictory with the standard result on SEG (Juditsky et al., 2011) saying that the average of the iterates computed by SEG does converge to the Nash equilibrium of the game. However, an important assumption made by Juditsky et al. is that the iterates are projected onto a compact set and that estimator of the gradient has finite variance. These assumptions break in this example since the variance of the estimator is proportional to the norm of the (unbounded) parameters. Note that constraining the optimization problem (23) to bounded domains $\Theta$ and $\Phi$, would make the finite variance assumption from Juditsky et al. (2011) holds. Consequently, the averaged iterate $\bar{\boldsymbol{\omega}}_t := \frac{1}{t} \sum_{s=0}^{t-1} \boldsymbol{\omega}_s$ would converge to $\boldsymbol{\omega}^*$. In § A.1, we explain why in a *non-convex setting*, the convergence of the *last iterate* is preferable.

# 3 Reducing Noise in Games with Variance Reduced Extragradient

One way to reduce the noise in the estimation of the gradient is to use mini-batches of samples instead of one sample. However, mini-batch stochastic extragradient fails to converge on (23) if the mini-batch size is smaller than half of the dataset size (see § C.1). In order to get an estimator of the gradient with a vanishing variance, the optimization literature proposed to take advantage of the finite-sum formulation that often appears in machine learning (Schmidt et al., 2017, and references therein).

## 3.1 Variance Reduced Gradient Methods

Let us assume that the objective in (2P-G) can be decomposed as a finite sum such that[5]

$$\mathcal{L}^G(\boldsymbol{\omega}) = \frac{1}{n} \sum_{i=1}^{n} \mathcal{L}_i^G(\boldsymbol{\omega}) \quad \text{and} \quad \mathcal{L}^D(\boldsymbol{\omega}) = \frac{1}{n} \sum_{i=1}^{n} \mathcal{L}_i^D(\boldsymbol{\omega}) \quad \text{where} \quad \boldsymbol{\omega} := (\boldsymbol{\theta}, \boldsymbol{\varphi}) \,. \quad (4)$$

Johnson and Zhang (2013) propose the "stochastic variance reduced gradient" (SVRG) as an *unbiased* estimator of the gradient with a smaller variance than the vanilla mini-batch estimate. The idea is to occasionally take a snapshot $\boldsymbol{\omega}^{\mathcal{S}}$ of the current model's parameters, and store the full batch gradient $\boldsymbol{\mu}^{\mathcal{S}}$ at this point. Computing the full batch gradient $\boldsymbol{\mu}^{\mathcal{S}}$ at $\boldsymbol{\omega}^{\mathcal{S}}$ is an expensive operation but not prohibitive if done infrequently (for instance once every dataset pass).

Assuming that we have stored $\boldsymbol{\omega}^{\mathcal{S}}$ and $\boldsymbol{\mu}^{\mathcal{S}} := (\boldsymbol{\mu}_{\boldsymbol{\theta}}^{\mathcal{S}}, \boldsymbol{\mu}_{\boldsymbol{\varphi}}^{\mathcal{S}})$, the *SVRG estimates* of the gradients are:

$$\boldsymbol{d}_i^G(\boldsymbol{\omega}) := \frac{\nabla \mathcal{L}_i^G(\boldsymbol{\omega}) - \nabla \mathcal{L}_i^G(\boldsymbol{\omega}^{\mathcal{S}})}{n \pi_i} + \boldsymbol{\mu}_{\boldsymbol{\theta}}^{\mathcal{S}} \,, \quad \boldsymbol{d}_i^D(\boldsymbol{\omega}) := \frac{\nabla \mathcal{L}_i^D(\boldsymbol{\omega}) - \nabla \mathcal{L}_i^D(\boldsymbol{\omega}^{\mathcal{S}})}{n \pi_i} + \boldsymbol{\mu}_{\boldsymbol{\varphi}}^{\mathcal{S}}. \quad (5)$$

These estimates are unbiased: $\mathbb{E}[\boldsymbol{d}_i^G(\boldsymbol{\omega})] = \frac{1}{n} \sum_{i=1}^{n} \nabla \mathcal{L}_i^G(\boldsymbol{\omega}) = \nabla \mathcal{L}^G(\boldsymbol{\omega})$, where the expectation is taken over $i$, picked with probability $\pi_i$. The non-uniform sampling probabilities $\pi_i$ are used to bias the sampling according to the Lipschitz constant of the stochastic gradient in order to sample more often gradients that change quickly. This strategy has been first introduced for variance reduced methods by Xiao and Zhang (2014) for SVRG and has been discussed for saddle point optimization by Palaniappan and Bach (2016).

Originally, SVRG was introduced as an epoch based algorithm with a *fixed epoch size*: in Alg. 1, one epoch is an inner loop of size $N$ (Line 6). However, Hofmann et al. (2015) proposed instead to

*sample* the size of each epoch from a geometric distribution, enabling them to analyze SVRG the same way as SAGA under a unified framework called $q$-memorization algorithm. We generalize their framework to handle the extrapolation step (EG) and provide a convergence proof for such $q$-memorization algorithms for games in § C.2.

One advantage of Hofmann et al. (2015)'s framework is also that the sampling of the epoch size does not depend on the condition number of the problem, whereas the original proof for SVRG had to consider an epoch size larger than the condition number (see Leblond et al. (2018, Corollary 16) for a detailed discussion on the convergence rate for SVRG). Thus, this new version of SVRG with a random epoch size becomes *adaptive to the local strong convexity* since none of its hyper-parameters depend on the strong convexity constant.

However, because of some new technical aspects when working with monotone operators, Palaniappan and Bach (2016)'s proofs (both for SAGA and SVRG) require a step-size (and epoch length for SVRG) that depends on the strong monotonicity constant making these algorithms not adaptive to local strong monotonicity. This motivates the proposed SVRE algorithm, which may be adaptive to local strong monotonicity, and is thus more appropriate for non-convex optimization.

## 3.2 SVRE: Stochastic Variance Reduced Extragradient

We describe our proposed algorithm called stochastic variance reduced extragradient (SVRE) in Alg. 1. In an analogous manner to how Palaniappan and Bach (2016) combined SVRG with the gradient method, SVRE combines SVRG estimates of the gradient (5) with the *extragradient method* (EG).

With SVRE we are able to improve the convergence rates for variance reduction for a large class of stochastic games (see Table 1 and Thm. 2), and we show in § 3.3 that it is the only method which empirically converges on the simple example of § 2.2.

We now describe the theoretical setup for the convergence result. A standard assumption in convex optimization is the assumption of strong convexity of the function. However, in a game, the operator,

$$v : \omega \mapsto \left[ \nabla_{\boldsymbol{\theta}} \mathcal{L}^G(\boldsymbol{\omega}), \ \nabla_{\boldsymbol{\varphi}} \mathcal{L}^D(\boldsymbol{\omega}) \right]^\top, \tag{6}$$

associated with the updates is no longer the gradient of a single function. To make an analogous assumption for games the optimization literature considers the notion of *strong monotonicity*.

**Definition 1.** *An operator $F : \boldsymbol{\omega} \mapsto (F_{\boldsymbol{\theta}}(\boldsymbol{\omega}), F_{\boldsymbol{\varphi}}(\boldsymbol{\omega})) \in \mathbb{R}^{d+p}$ is said to be $(\mu_{\boldsymbol{\theta}}, \mu_{\boldsymbol{\varphi}})$-strongly monotone if for all $\boldsymbol{\omega}, \boldsymbol{\omega}' \in \mathbb{R}^{p+d}$ we have*

$$\Omega((\boldsymbol{\theta}, \boldsymbol{\varphi}), (\boldsymbol{\theta}', \boldsymbol{\varphi}')) := \mu_{\boldsymbol{\theta}} \|\boldsymbol{\theta} - \boldsymbol{\theta}'\|^2 + \mu_{\boldsymbol{\varphi}} \|\boldsymbol{\varphi} - \boldsymbol{\varphi}'\|^2 \leq (F(\boldsymbol{\omega}) - F(\boldsymbol{\omega}'))^\top (\boldsymbol{\omega} - \boldsymbol{\omega}'),$$

*where we write $\boldsymbol{\omega} := (\boldsymbol{\theta}, \boldsymbol{\varphi}) \in \mathbb{R}^{d+p}$. A monotone operator is a $(0, 0)$-strongly monotone operator.*

This definition is a generalization of strong convexity for operators: if $f$ is $\mu$-strongly convex, then $\nabla f$ is a $\mu$-monotone operator. Another assumption is the $\gamma$ regularity assumption,

**Definition 2.** *An operator $F : \boldsymbol{\omega} \mapsto (F_{\boldsymbol{\theta}}(\boldsymbol{\omega}), F_{\boldsymbol{\varphi}}(\boldsymbol{\omega})) \in \mathbb{R}^{d+p}$ is said to be $(\gamma_{\boldsymbol{\theta}}, \gamma_{\phi})$-regular if,*

$$\gamma_{\boldsymbol{\theta}}^2 \|\boldsymbol{\theta} - \boldsymbol{\theta}'\|^2 + \gamma_{\boldsymbol{\varphi}}^2 \|\boldsymbol{\varphi} - \boldsymbol{\varphi}'\|^2 \leq \|F(\boldsymbol{\omega}) - F(\boldsymbol{\omega}')\|^2, \quad \forall \boldsymbol{\omega}, \boldsymbol{\omega}' \in \mathbb{R}^{p+d}. \tag{7}$$

Note that an *operator* is always $(0, 0)$-regular. This assumption originally introduced by Tseng (1995) has been recently used (Azizian et al., 2019) to improve the convergence rate of extragradient. For instance for a full rank bilinear matrix problem $\gamma$ is its smallest singular value. More generally, in the case $\gamma_{\boldsymbol{\theta}} = \gamma_{\boldsymbol{\varphi}}$, the regularity constant is a lower bound on the minimal singular value of the Jacobian of $F$ (Azizian et al., 2019).

One of our main assumptions is the cocoercivity assumption, which implies the Lipchitzness of the operator in the unconstrained case. We use the cocoercivity constant because it provides a tighter bound for general strongly monotone and Lipschitz games (see discussion following Theorem 2).

**Definition 3.** *An operator $F : \boldsymbol{\omega} \mapsto (F_{\boldsymbol{\theta}}(\boldsymbol{\omega}), F_{\boldsymbol{\varphi}}(\boldsymbol{\omega})) \in \mathbb{R}^{d+p}$ is said to be $(\ell_{\boldsymbol{\theta}}, \ell_{\boldsymbol{\varphi}})$-cocoercive, if for all $\boldsymbol{\omega}, \boldsymbol{\omega}' \in \Omega$ we have*

$$\|F(\boldsymbol{\omega}) - F(\boldsymbol{\omega}')\|^2 \leq \ell_{\boldsymbol{\theta}} (F_{\boldsymbol{\theta}}(\boldsymbol{\omega}) - F_{\boldsymbol{\theta}}(\boldsymbol{\omega}'))^\top (\boldsymbol{\theta} - \boldsymbol{\theta}') + \ell_{\boldsymbol{\varphi}} (F_{\boldsymbol{\varphi}}(\boldsymbol{\omega}) - F_{\boldsymbol{\varphi}}(\boldsymbol{\omega}'))^\top (\boldsymbol{\varphi} - \boldsymbol{\varphi}'). \tag{8}$$

Note that for a $L$-Lipschitz and $\mu$-strongly monotone operator, we have $\ell \in [L, L^2/\mu]$ (Facchinei and Pang, 2003). For instance, when $F$ is the gradient of a convex function, we have $\ell = L$. More generally, when $F(\boldsymbol{\omega}) = (\nabla f(\boldsymbol{\theta}) + M\boldsymbol{\varphi}, \nabla g(\boldsymbol{\varphi}) - M^\top \boldsymbol{\theta})$, where $f$ and $g$ are $\mu$-strongly convex and $L$ smooth we have that $\gamma = \sigma_{\min}(M)$ and $\|M\|^2 = O(\mu L)$ is a sufficient condition for $\ell = O(L)$ (see §B). Under this assumption on each cost function of the game operator, we can define a cocoercivity constant adapted to the non-uniform sampling scheme of our stochastic algorithm:

$$\bar{\ell}(\pi)^2 := \frac{1}{n} \sum_{i=1}^n \frac{1}{n\pi_i} \ell_i^2. \tag{9}$$

The standard *uniform sampling scheme* corresponds to $\pi_i := \frac{1}{n}$ and the optimal *non-uniform* sampling scheme corresponds to $\tilde{\pi}_i := \frac{\ell_i}{\sum_{i=1}^n \ell_i}$. By Jensen's inequality, we have: $\bar{\ell}(\tilde{\pi}) \leq \bar{\ell}(\pi) \leq \max_i \ell_i$.

For our main result, we make strong convexity, cocoercivity and regularity assumptions.

**Assumption 1.** *For $1 \leq i \leq n$, the gradients $\nabla_{\boldsymbol{\theta}} \mathcal{L}_i^G$ and $\nabla_{\boldsymbol{\varphi}} \mathcal{L}_i^D$ are respectively $\ell_i^{\boldsymbol{\theta}}$ and $\ell_i^{\boldsymbol{\varphi}}$-cocoercive and $(\gamma_i^{\boldsymbol{\theta}}, \gamma_i^{\boldsymbol{\varphi}})$-regular. The operator (6) is $(\mu_{\boldsymbol{\theta}}, \mu_{\boldsymbol{\varphi}})$-strongly monotone.*

We now present our convergence result for SVRE with non-uniform sampling (to make our constants comparable to those of Palaniappan and Bach (2016)), but note that we have used uniform sampling in all our experiments (for simplicity).

**Theorem 2.** *Under Assumption 1, after $t$ iterations, the iterate $\boldsymbol{\omega}_t := (\boldsymbol{\theta}_t, \boldsymbol{\varphi}_t)$ computed by SVRE (Alg. 1) with step-size $\eta_{\boldsymbol{\theta}} \leq (40\bar{\ell}_{\boldsymbol{\theta}})^{-1}$ and $\eta_{\boldsymbol{\varphi}} \leq (40\bar{\ell}_{\boldsymbol{\varphi}})^{-1}$ and sampling scheme $(\tilde{\pi}_{\boldsymbol{\theta}}, \tilde{\pi}_{\boldsymbol{\varphi}})$ verifies:*

$$\mathbb{E}[\|\boldsymbol{\omega}_t - \boldsymbol{\omega}^*\|_2^2] \leq \left(1 - \frac{1}{2}\min\left\{\eta_{\boldsymbol{\theta}}\mu_{\boldsymbol{\theta}} + \frac{9\eta_{\boldsymbol{\theta}}^2\bar{\gamma}_{\boldsymbol{\theta}}^2}{10}, \eta_{\boldsymbol{\varphi}}\mu_{\boldsymbol{\varphi}} + \frac{9\eta_{\boldsymbol{\varphi}}^2\bar{\gamma}_{\boldsymbol{\varphi}}^2}{10}, \frac{4}{5n}\right\}\right)^t \mathbb{E}[\|\boldsymbol{\omega}_0 - \boldsymbol{\omega}^*\|_2^2],$$

*where $\bar{\ell}_{\boldsymbol{\theta}}(\pi_{\boldsymbol{\theta}})$ and $\bar{\ell}_{\boldsymbol{\varphi}}(\pi_{\boldsymbol{\varphi}})$ are defined in (9). Particularly, for $\eta_{\boldsymbol{\theta}} = \frac{1}{40\bar{\ell}_{\boldsymbol{\theta}}}$ and $\eta_{\boldsymbol{\varphi}} = \frac{1}{40\bar{\ell}_{\boldsymbol{\varphi}}}$ we get*

$$\mathbb{E}[\|\boldsymbol{\omega}_t - \boldsymbol{\omega}^*\|_2^2] \leq \left(1 - \frac{1}{2}\min\left\{\frac{1}{40}\left(\frac{\mu_{\boldsymbol{\theta}}}{\bar{\ell}_{\boldsymbol{\theta}}} + \frac{\bar{\gamma}_{\boldsymbol{\theta}}^2}{45\bar{\ell}_{\boldsymbol{\theta}}^2}\right), \frac{1}{40}\left(\frac{\mu_{\boldsymbol{\varphi}}}{\bar{\ell}_{\boldsymbol{\varphi}}} + \frac{\bar{\gamma}_{\boldsymbol{\varphi}}^2}{45\bar{\ell}_{\boldsymbol{\varphi}}^2}\right), \frac{4}{5n}\right\}\right)^t \mathbb{E}[\|\boldsymbol{\omega}_0 - \boldsymbol{\omega}^*\|_2^2].$$

We prove this theorem in § C.2. We can notice that the respective *condition numbers* of $\mathcal{L}^G$ and $\mathcal{L}^D$ defined as $\kappa_{\boldsymbol{\theta}} := \frac{\mu_{\boldsymbol{\theta}}}{\bar{\ell}_{\boldsymbol{\theta}}} + \frac{\bar{\gamma}_{\boldsymbol{\theta}}^2}{\bar{\ell}_{\boldsymbol{\theta}}^2}$ and $\kappa_{\boldsymbol{\varphi}} := \frac{\mu_{\boldsymbol{\varphi}}}{\bar{\ell}_{\boldsymbol{\varphi}}} + \frac{\bar{\gamma}_{\boldsymbol{\varphi}}^2}{\bar{\ell}_{\boldsymbol{\varphi}}^2}$ appear in our convergence rate. The cocoercivity constant $\ell$ belongs to $[L, L^2/\mu]$, thus our rate may be significantly faster[6] than the convergence rate of the (non-accelerated) algorithm of Palaniappan and Bach (2016) that depends on the product $\frac{\mu_{\boldsymbol{\theta}}}{L_{\boldsymbol{\theta}}}\frac{\mu_{\boldsymbol{\varphi}}}{L_{\boldsymbol{\varphi}}}$. They avoid a dependence on the maximum of the condition numbers squared, $\max\{\kappa_{\boldsymbol{\varphi}}^2, \kappa_{\boldsymbol{\theta}}^2\}$, by using the weighted Euclidean norm $\Omega(\boldsymbol{\theta}, \boldsymbol{\varphi})$ defined in (14) and rescaling the functions $\mathcal{L}^G$ and $\mathcal{L}^D$ with their strong-monotonicity constant. However, this rescaling trick suffers from two issues: (i) we do not know in practice a good estimate of the strong monotonicity constant, which was *not* the case in Palaniappan and Bach (2016)'s application; and (ii) the algorithm does not adapt to local strong-monotonicity. This property is important in non-convex optimization since we want the algorithm to exploit the (potential) local stability properties of a stationary point.

### 3.3 Motivating example

The example (23) for $\epsilon = 0$ seems to be challenging in the stochastic setting since all the standard methods and even the stochastic extragradient method fails to find its Nash equilibrium (note that this example is *not* strongly monotone). We set $n = d = 100$, and draw $[\boldsymbol{A}_i]_{kl} = \delta_{kli}$ and $[\boldsymbol{b}_i]_k, [\boldsymbol{c}_i]_k \sim \mathcal{N}(0, 1/d)$, $1 \leq k, l \leq d$, where $\delta_{kli} = 1$ if $k = l = i$ and 0 otherwise. Our optimization problem is:

$$\min_{\boldsymbol{\theta} \in \mathbb{R}^d} \max_{\boldsymbol{\varphi} \in \mathbb{R}^d} \frac{1}{n} \sum_{i=1}^n (\boldsymbol{\theta}^\top \boldsymbol{b}_i + \boldsymbol{\theta}^\top \boldsymbol{A}_i \boldsymbol{\varphi} + \boldsymbol{c}_i^\top \boldsymbol{\varphi}). \tag{10}$$

We compare variants of the following algorithms (with uniform sampling and average our results over 5 different seeds): (i) AltSGD: the standard method to train GANs–stochastic gradient with alternating updates of each player. (ii) SVRE: Alg. 1. The AVG prefix correspond to the *uniform average* of the iterates, $\bar{\omega} := \frac{1}{t}\sum_{s=0}^{t-1}\omega_s$. We observe in Fig. 4 that AVG-SVRE converges sublinearly (whereas AVG-AltSGD fails to converge).

This motivates a new variant of SVRE based on the idea that even if the averaged iterate converges, we do not compute the gradient at that point and thus we do not benefit from the fact that this iterate is closer to the optimums (see § A.1). Thus the idea is to occasionally restart the algorithm, i.e., consider the averaged iterate as the new starting point of our algorithm and compute the gradient at that point. Restart goes well with SVRE as we already occasionally stop the inner loop to recompute $\mu^{\mathcal{S}}$, at which point we decide (with a probability $p$ to be fixed) whether or not to restart the algorithm by taking the snapshot at point $\bar{\omega}_t$ instead of $\omega_t$. This variant of SVRE is described in Alg. 3 in § E and the variant combining VRAd in § D.1.

In Fig. 4 we observe that the only method that converges is SVRE and its variants. We do not provide convergence guarantees for Alg. 3 and leave its analysis for future work. However, it is interesting that, to our knowledge, this algorithm is the only stochastic algorithm (excluding batch extragradient as it is not stochastic) that converge for (23). Note that we tried all the algorithms presented in Fig. 3 from Gidel et al. (2019a) on this *unconstrained* problem and that all of them diverge.

## 4    GAN Experiments

In this section, we investigate the empirical performance of SVRE for *GAN training*. Note, however, that our theoretical analysis does not hold for games with non-convex objectives such as GANs.

**Datasets.**    We used the following datasets: (i) **MNIST** (Lecun and Cortes), (ii) **CIFAR-10** (Krizhevsky, 2009, §3), (iii) **SVHN** (Netzer et al., 2011), and (iv) **ImageNet** ILSVRC 2012 (Russakovsky et al., 2015), using $28\times28$, $3\times32\times32$, $3\times32\times32$, and $3\times64\times64$ resolution, respectively.

**Metrics.**    We used the **Inception score** (IS, Salimans et al., 2016) and the **Fréchet Inception distance** (FID, Heusel et al., 2017) as performance metrics for image synthesis. To gain insights if SVRE indeed reduces the variance of the gradient estimates, we used the **second moment estimate– SME** (uncentered variance), computed with an exponentially moving average. See § F.1 for details.

**DNN architectures.**    For experiments on **MNIST**, we used the DCGAN architectures (Radford et al., 2016), described in § F.2.1. For real-world datasets, we used two architectures (see § F.2 for details and § F.2.2 for motivation): (i) SAGAN (Zhang et al., 2018), and (ii) ResNet, replicating the setup of Miyato et al. (2018), described in detail in § F.2.3 and F.2.4, respectively. For clarity, we refer the former as *shallow*, and the latter as *deep* architectures.

**Optimization methods.**    We conduct experiments using the following optimization methods for GANs: (i) **BatchE:** full–batch extragradient, (ii) **SG:** stochastic gradient (alternating GAN), and (iii) **SE:** stochastic extragradient, and (iv) **SVRE:** stochastic variance reduced extragradient. These can be combined with adaptive learning rate methods such as *Adam* or with parameter averaging, hereafter denoted as **–A** and **AVG–**, respectively. In § D.1, we present a variant of Adam adapted to variance reduced algorithms, that is referred to as **–VRAd**. When using the SE–A baseline and *deep* architectures, the convergence rapidly fails at some point of training (cf. § G.3). This motivates experiments where we start from a stored checkpoint taken *before* the baseline diverged, and *continue training with SVRE*. We denote these experiments with **WS–SVRE** (warm-start SVRE).

### 4.1    Results

**Comparison on MNIST.**    The **MNIST** common benchmark allowed for comparison with full-batch extragradient, as it is feasible to compute. Fig. 3 depicts the IS metric while using either a stochastic, full-batch or variance reduced version of extragradient (see details of SVRE-GAN in § D.2). We always combine the stochastic baseline (SE) with *Adam*, as proposed by Gidel et al. (2019a). In terms of number of parameter updates, SVRE performs similarly to BatchE–A (see Fig. 5a, § G). Note that the latter requires significantly more computation: Fig. 3a depicts the IS metric using the number of mini-batch computations as x-axis (a surrogate for the wall-clock time, see below). We observe that,

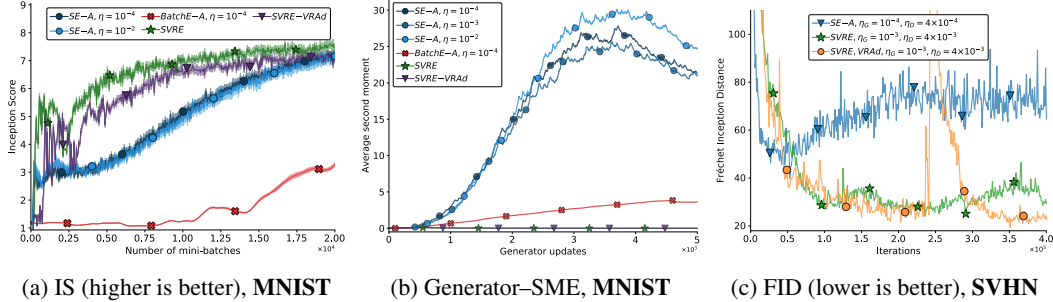

(a) IS (higher is better), **MNIST**    (b) Generator–SME, **MNIST**    (c) FID (lower is better), **SVHN**

Figure 3: **Figures a & b.** Stochastic, full-batch and variance reduced extragradient optimization on **MNIST**. We used $\eta = 10^{-2}$ for SVRE. *SE–A* with $\eta = 10^{-3}$ achieves similar IS performances as $\eta = 10^{-2}$ and $\eta = 10^{-4}$, omitted from Fig. a for clarity. **Figure c.** FID on **SVHN**, using *shallow* architectures. See § 4 and § F for naming of methods and details on the implementation, respectively.

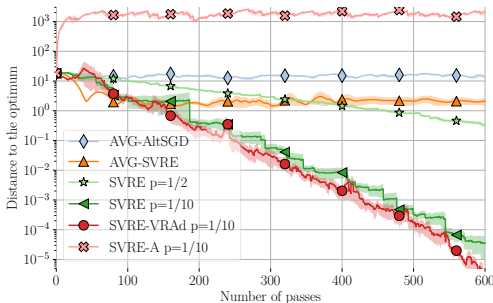

Figure 4: Distance to the optimum of (10), see § 3.3 for the experimental setup.

|  | SG-A | SE-A | SVRE | WS-SVRE |
|---|---|---|---|---|
| CIFAR-10 | 21.70 | 18.65 | 23.56 | **16.77** |
| SVHN | 5.66 | 5.14 | **4.81** | 4.88 |

Table 2: Best obtained FID scores for the different optimization methods using the *deep* architectures (see Table 8, § F.2.4). WS–SVRE starts from the best obtained scores of SE–A. See § F and § G for implementation details and additional results, respectively.

as SE–A has slower per-iteration convergence rate, SVRE converges faster on this dataset. At the end of training, all methods reach similar performances (IS is above 8.5, see Table 9, § G).

**Computational cost.** The relative cost of one pass over the dataset for SVRE versus vanilla SGD is a factor of 5: the full batch gradient is computed (on average) after one pass over the dataset, giving a slowdown of 2; the factor 5 takes into account the extra stochastic gradient computations for the variance reduction, as well as the extrapolation step overhead. However, as SVRE provides less noisy gradient, it may converge faster per iteration, compensating the extra per-update cost. Note that many computations can be done in parallel. In Fig. 3a, the x-axis uses an implementation-independent surrogate for wall-clock time that counts the number of mini-batch gradient computations. Note that some training methods for GANs require multiple discriminator updates per generator update, and we observed that to stabilize our baseline when using the *deep* architectures it was required to use 1:5 update ratio of $G:D$ (cf. § G.3), whereas for SVRE we used ratio of 1:1 (Tab. 2 lists the results).

**Second moment estimate and Adam.** Fig. 3b depicts the averaged second-moment estimate for parameters of the Generator, where we observe that SVRE effectively reduces it over the iterations. The reduction of these values may be the reason why Adam combined with SVRE performs poorly (as these values appear in the denominator, see § D.1). To our knowledge, SVRE is the first optimization method with a constant step size that has worked empirically for GANs on non-trivial datasets.

**Comparison on real-world datasets.** In Fig. 3c, we compare SVRE with the SE–A baseline on **SVHN**, using *shallow* architectures. We observe that although SE–A in some experiments obtains better performances in the early iterations, SVRE allows for obtaining improved *final* performances. Tab. 2 summarizes the results on **CIFAR-10** and **SVHN** with *deep* architectures. We observe that, with deeper architectures, SE–A is notably more unstable, as training collapsed in 100% of the experiments. To obtain satisfying results for SE–A, we used various techniques such as a schedule of the learning rate and different update ratios (see § G.3). On the other hand, SVRE *did not collapse in any of the experiments* but took longer time to converge compared to SE–A. Interestingly, although

WS–SVRE starts from an iterate point after which the baseline diverges, it continues to improve the obtained FID score and does not diverge. See § G for additional experiments.

## 5  Related work

Surprisingly, there exist only a few works on variance reduction methods for monotone operators, namely from Palaniappan and Bach (2016) and Davis (2016). The latter requires a co-coercivity assumption on the operator and thus only convex optimization is considered. Our work provides a new way to use variance reduction for monotone operators, using the extragradient method (Korpelevich, 1976). Recently, Iusem et al. (2017) proposed an extragradient method with variance reduction for an *infinite sum* of operators. The authors use mini-batches of growing size in order to reduce the variance of their algorithm and to converge with a constant step-size. However, this approach is prohibitively expensive in our application. Moreover, Iusem et al. are not using the SAGA/SVRG style of updates exploiting the finite sum formulation, leading to sublinear convergence rate, while our method benefits from a linear convergence rate exploiting the finite sum assumption.

Daskalakis et al. (2018) proposed a method called Optimistic-Adam inspired by game theory. This method is closely related to extragradient, with slightly different update scheme. More recently, Gidel et al. (2019a) proposed to use extragradient to train GANs, introducing a method called ExtraAdam. This method outperformed Optimistic-Adam when trained on CIFAR-10. Our work is also an attempt to find principled ways to train GANs. Considering that the game aspect is better handled by the extragradient method, we focus on the optimization issues arising from the noise in the training procedure, a disregarded potential issue in GAN training.

In the context of deep learning, despite some very interesting theoretical results on non-convex minimization (Reddi et al., 2016; Allen-Zhu and Hazan, 2016), the effectiveness of variance reduced methods is still an open question, and a recent technical report by Defazio and Bottou (2018) provides negative empirical results on the variance reduction aspect. In addition, two recent large scale studies showed that increased batch size has: (i) only marginal impact on *single objective training* (Shallue et al., 2018) and (ii) a surprisingly large performance improvement on *GAN training* (Brock et al., 2019). In our work, we are able to show positive results for variance reduction in a real-world deep learning setting. This unexpected difference seems to confirm the remarkable discrepancy, that remains poorly understood, between multi-objective optimization and standard minimization.

## 6  Discussion

Motivated by a simple bilinear game optimization problem where stochasticity provably breaks the convergence of previous stochastic methods, we proposed the novel SVRE algorithm that combines SVRG with the extragradient method for optimizing games. On the theory side, SVRE improves upon the previous best results for strongly-convex games, whereas empirically, it is the only method that converges for our stochastic bilinear game counter-example.

We empirically observed that SVRE for GAN training obtained convergence speed similar to Batch-Extragradient on MNIST, while the latter is computationally infeasible for large datasets. For shallow architectures, SVRE matched or improved over baselines on all four datasets. Our experiments with deeper architectures show that SVRE is notably more stable with respect to hyperparameter choice. Moreover, while its stochastic counterpart diverged in all our experiments, SVRE did not. However, we observed that SVRE took more iterations to converge when using deeper architectures, though notably, we were using constant step-sizes, unlike the baselines which required Adam. As adaptive step-sizes often provide significant improvements, developing such an appropriate version for SVRE is a promising direction for future work. In the meantime, the stability of SVRE suggests a practical use case for GANs as warm-starting it just before the baseline diverges, and running it for further improvements, as demonstrated with the WS–SVRE method in our experiments.

## Acknowledgements

This research was partially supported by the Canada CIFAR AI Chair Program, the Canada Excellence Research Chair in "Data Science for Realtime Decision-making", by the NSERC Discovery Grant RGPIN-2017-06936, by the Hasler Foundation through the MEMUDE project, and by a Google

Focused Research Award. Authors would like to thank Compute Canada for providing the GPUs used for this research. TC would like to thank Sebastian Stich and Martin Jaggi, and GG and TC would like to thank Hugo Berard for helpful discussions.

## Footnotes

[3]For simplicity, we focus on *unconstrained* setting where $\Theta = \mathbb{R}^d$. For the *constrained* case, a Euclidean projection on the constraints set should be added at every update of the method.

[4]Gidel et al. (2019b) formalize the notion of "adversarial component" of a game, which yields a rotational dynamics in gradients methods (oscillations in parameters), as illustrated by the gradient field of Fig. 1 (right).

[5]The "noise dataset" in a GAN is not finite though; see § D.1 for details on how to cope with this in practice.

[6]Particularly, when $F$ is the gradient of a convex function (or close to it) we have $\ell \approx L$ and thus our rate recovers the standard $\ln(1/\epsilon)L/\mu$, improving over the accelerated algorithm of Palaniappan and Bach (2016). More generally, under the assumptions of Proposition 2, we also recover $\ln(1/\epsilon)L/\mu$.

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
