[Supplementary Material · svre_full.pdf]

# Reducing Noise in GAN Training with Variance Reduced Extragradient

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

**Algorithm 1** Pseudocode for SVRE.

1: **Input:** Stopping time $T$, learning rates $\eta_{\theta}, \eta_{\varphi}$, initial weights $\theta_0, \varphi_0$. $t = 0$
2: **while** $t \le T$ **do**
3: $\quad \varphi^{\mathcal{S}} = \varphi_t$ and $\mu^{\mathcal{S}}_{\varphi} = \frac{1}{n}\sum_{i=1}^{n}\nabla_{\varphi}\mathcal{L}_i^D(\theta^{\mathcal{S}},\varphi^{\mathcal{S}})$
4: $\quad \theta^{\mathcal{S}} = \theta_t$ and $\mu^{\mathcal{S}}_{\theta} = \frac{1}{n}\sum_{i=1}^{n}\nabla_{\theta}\mathcal{L}_i^G(\theta^{\mathcal{S}},\varphi^{\mathcal{S}})$
5: $\quad N \sim \mathrm{Geom}\left(1/n\right)$ *(Sample epoch length)*
6: $\quad$**for** $i = 0$ **to** $N-1$ **do** {*Beginning of the epoch*}
7: $\quad\quad$**Sample** $i_{\theta}, i_{\varphi} \sim \pi_{\theta}, \pi_{\varphi}$, do **extrapolation:**
8: $\quad\quad \tilde{\varphi}_t = \varphi_t - \eta_{\varphi} d_{i_{\varphi}}^D(\theta_t,\varphi_t,\theta^{\mathcal{S}},\varphi^{\mathcal{S}})$ $\quad\triangleright$ (5)
9: $\quad\quad \tilde{\theta}_t = \theta_t - \eta_{\theta} d_{i_{\theta}}^G(\theta_t,\varphi_t,\theta^{\mathcal{S}},\varphi^{\mathcal{S}})$ $\quad\triangleright$ (5)
10: $\quad\quad$**Sample** $i_{\theta}, i_{\varphi} \sim \pi_{\theta}, \pi_{\varphi}$ and do **update:**
11: $\quad\quad \varphi_{t+1} = \varphi_t - \eta_{\varphi} d_{i_{\varphi}}^D(\tilde{\theta}_t,\tilde{\varphi}_t,\theta^{\mathcal{S}},\varphi^{\mathcal{S}})$ $\triangleright$ (5)
12: $\quad\quad \theta_{t+1} = \theta_t - \eta_{\theta} d_{i_{\theta}}^G(\tilde{\theta}_t,\tilde{\varphi}_t,\theta^{\mathcal{S}},\varphi^{\mathcal{S}})$ $\quad\triangleright$ (5)
13: $\quad\quad t \leftarrow t + 1$
14: **Output:** $\theta_T, \varphi_T$

The theoretical aspect we present in this paper is further supported empirically, since using larger mini-batch sizes for GAN training has been shown to considerably improve the quality of the samples produced by the resulting generative model: Brock et al. (2019) report a relative improvement of $46\%$ of the Inception Score metric (see § 4) on ImageNet if the batch size is increased 8–fold. This notable improvement raises the question if noise reduction optimization methods can be extended to game settings. In turn, this would allow for a principled training method with the practical benefit of omitting to empirically establish this multiplicative factor for the batch size.

In this paper, we investigate the interplay between noise and multi-objective problems in the context of GAN training. Our contributions can be summarized as follows: (i) we show in a motivating example how the noise can make stochastic extragradient fail (see § 2.2). (ii) we propose a new method "stochastic variance reduced extragradient" (SVRE) that combines variance reduction and extrapolation (see Alg. 1 and § 3.2) and show experimentally that it effectively reduces the noise. (iii) we prove the convergence of SVRE under local strong convexity assumptions, improving over the known rates of competitive methods for a large class of games (see § 3.2 for our convergence result and Table 1 for comparison with standard methods). (iv) we test SVRE empirically to train GANs on several standard datasets, and observe that it can improve SOTA deep models in the late stage of their optimization (see § 4).

## 2 GANs as a Game and Noise in Games

### 2.1 Game theory formulation of GANs

The models in a GAN are a generator $G$, that maps an embedding space to the signal space, and should eventually map a fixed noise distribution to the training data distribution, and a discriminator $D$ whose purpose is to allow the training of the generator by classifying genuine samples against generated ones. At each iteration of the algorithm, the discriminator $D$ is updated to improve its "real vs. generated" classification performance, and the generator $G$ to degrade it.

From a game theory point of view, GAN training is a differentiable two-player game where the generator $G_{\theta}$ and the discriminator $D_{\varphi}$ aim at minimizing their own cost function $\mathcal{L}^G$ and $\mathcal{L}^D$, resp.:

$$\theta^* \in \arg\min_{\theta\in\Theta} \mathcal{L}^G(\theta,\varphi^*) \qquad \text{and} \qquad \varphi^* \in \arg\min_{\varphi\in\Phi} \mathcal{L}^D(\theta^*,\varphi). \qquad \text{(

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

[7]For instance, Daskalakis et al. (2018); Gidel et al. (2019a) plugged Adam into their principled method to get better results.

[8] https://pytorch.org/

[9] https://www.tensorflow.org/

[10] https://github.com/openai/improved-gan/

[11]https://github.com/bioinf-jku/TTUR

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

## A  Noise in games

### A.1  Why is convergence of the last iterate preferable?

In light of Theorem 1, the behavior of the iterates on the unconstrained version of (23) ($\epsilon = 0$):

$$\min_{\boldsymbol{\theta} \in \Theta} \max_{\boldsymbol{\varphi} \in \Phi} \frac{1}{n} \sum_{i=1}^{n} \boldsymbol{\theta}^{\top} \boldsymbol{A}_i \boldsymbol{\varphi} \quad \text{where} \quad [\boldsymbol{A}_i]_{kl} = 1 \text{ if } k = l = i \text{ and } 0 \text{ otherwise.} \tag{11}$$

where $\Theta$ and $\Phi$ are compact and convex sets, is the following: they will diverge until they reach the boundary of $\Theta$ and $\Phi$ and then they will start to turn around the Nash equilibrium of (11) lying on these boundaries. Using convexity properties, we can then show that the averaged iterates will converge to the Nash equilibrium of the problem. However, with an arbitrary large domain, this convergence rate may be arbitrary slow (since it depends on the diameter of the domain).

Moreover, this behavior might be even more problematic in a non-convex framework because even if by chance we initialize close to the Nash equilibrium, we would get away from it and we cannot rely on convexity to expect the average of the iterates to converge.

Consequently, we would like optimization algorithms generating iterates that *stay close to the Nash equilibrium.*

## B  Definitions and Lemmas

### B.1  Smoothness and Monotonicity of the operator

Another important property used is the Lipschitzness of an operator.

**Definition 4.** *A mapping $F : \mathbb{R}^p \to \mathbb{R}^d$ is said to be L-Lipschitz if,*

$$\|F(\boldsymbol{\omega}) - F(\boldsymbol{\omega}')\|_2 \leq L\|\boldsymbol{\omega} - \boldsymbol{\omega}'\|_2, \quad \forall \boldsymbol{\omega}, \boldsymbol{\omega}' \in \Omega. \tag{12}$$

**Definition 5.** *A differentiable function $f : \Omega \to \mathbb{R}$ is said to be $\mu$-strongly convex if*

$$f(\boldsymbol{\omega}) \geq f(\boldsymbol{\omega}') + \nabla f(\boldsymbol{\omega}')^{\top}(\boldsymbol{\omega} - \boldsymbol{\omega}') + \frac{\mu}{2}\|\boldsymbol{\omega} - \boldsymbol{\omega}'\|_2^2 \quad \forall \boldsymbol{\omega}, \boldsymbol{\omega}' \in \Omega. \tag{13}$$

**Definition 6.** *A function $(\boldsymbol{\theta}, \boldsymbol{\varphi}) \mapsto \mathcal{L}(\boldsymbol{\theta}, \boldsymbol{\varphi})$ is said convex-concave if $\mathcal{L}(\cdot, \boldsymbol{\varphi})$ is convex for all $\boldsymbol{\varphi} \in \Phi$ and $\mathcal{L}(\boldsymbol{\theta}, \cdot)$ is concave for all $\boldsymbol{\theta} \in \Theta$. An $\mathcal{L}$ is said to be $\mu$-strongly convex concave if $(\boldsymbol{\theta}, \boldsymbol{\varphi}) \mapsto \mathcal{L}(\boldsymbol{\theta}, \boldsymbol{\varphi}) - \frac{\mu}{2}\|\boldsymbol{\theta}\|_2^2 + \frac{\mu}{2}\|\boldsymbol{\varphi}\|_2^2$ is convex concave.*

**Definition 7.** *For $\mu_{\boldsymbol{\theta}}, \mu_{\boldsymbol{\varphi}} > 0$, an operator $F : \boldsymbol{\omega} \mapsto (F_{\boldsymbol{\theta}}(\boldsymbol{\omega}), F_{\boldsymbol{\varphi}}(\boldsymbol{\omega})) \in \mathbb{R}^{d+p}$ is said to be $(\mu_{\boldsymbol{\theta}}, \mu_{\boldsymbol{\varphi}})$-strongly monotone if $\forall \boldsymbol{\omega}, \boldsymbol{\omega}' \in \Omega \subset \mathbb{R}^{p+d}$,*

$$(F(\boldsymbol{\omega}) - F(\boldsymbol{\omega}'))^{\top}(\boldsymbol{\omega} - \boldsymbol{\omega}') \geq \mu_{\boldsymbol{\theta}}\|\boldsymbol{\theta} - \boldsymbol{\theta}'\|^2 + \mu_{\boldsymbol{\varphi}}\|\boldsymbol{\varphi} - \boldsymbol{\varphi}'\|^2.$$

*where we noted $\boldsymbol{\omega} := (\boldsymbol{\theta}, \boldsymbol{\varphi}) \in \mathbb{R}^{d+p}$.*

**Definition 8.** *An operator $F : (\boldsymbol{\omega}), \in \mathbb{R}^d$ is said to be $\ell$-cocoercive, if for all $\boldsymbol{\omega}, \boldsymbol{\omega}' \in \Omega$ we have*

$$\|F(\boldsymbol{\omega}) - F(\boldsymbol{\omega}')\|^2 \leq \ell(F(\boldsymbol{\omega}) - F(\boldsymbol{\omega}'))^{\top}(\boldsymbol{\omega} - \boldsymbol{\omega}'). \tag{14}$$

**Proposition 1** (Folklore). *A L-Lipschitz and $\mu$-strongly monotone operator is $L^2/\mu$-cocoercive*

*Proof.* By applying lipschitzness and strong monotonicity,

$$\|F(\boldsymbol{\omega}) - F(\boldsymbol{\omega}')\|^2 \leq L^2\|\boldsymbol{\omega} - \boldsymbol{\omega}'\|^2 \leq L^2/\mu(F(\boldsymbol{\omega}) - F(\boldsymbol{\omega}'))^{\top}(\boldsymbol{\omega} - \boldsymbol{\omega}') \tag{15}$$

$\square$

**Proposition 2.** *If $F(\boldsymbol{\omega}) = (\nabla f(\boldsymbol{\theta}) + M\boldsymbol{\varphi}, \nabla g(\boldsymbol{\varphi}) - M^\top \boldsymbol{\theta})$, where $f$ and $g$ are $\mu$-strongly convex and $L$ smooth, then $\|M\|^2 = O(\mu L)$ is a sufficient condition for $F$ to be $\ell$-cocoercive with $\ell = O(L)$*

*Proof.* We rewrite $F$ as the sum of the gradient of convex Lipschitz function $F_{grad}$ and a $L$-Lipschitz and $\mu$-strongly monotone operator $F_{mon}$:

$$F_{grad}(\boldsymbol{\omega}) := (\nabla f(\boldsymbol{\theta}) - \mu\boldsymbol{\theta}, \nabla g(\boldsymbol{\varphi}) - \mu\boldsymbol{\varphi}) \quad \text{and} \quad F_{mon} : (M\boldsymbol{\varphi} + \mu\boldsymbol{\theta}, -M^\top\boldsymbol{\theta} + \mu\boldsymbol{\varphi}) \tag{16}$$

Then

$$\|F(\boldsymbol{\omega}) - F(\boldsymbol{\omega}')\|^2 \leq 2\|F_{grad}(\boldsymbol{\omega}) - F_{grad}(\boldsymbol{\omega}')\|^2 + 2\|F_{mon}(\boldsymbol{\omega}) - F_{mon}(\boldsymbol{\omega}')\|^2 \tag{17}$$

$$\leq 2(L+\mu)(F_{grad}(\boldsymbol{\omega}) - F_{grad}(\boldsymbol{\omega}'))^\top(\boldsymbol{\omega} - \boldsymbol{\omega}') \tag{18}$$

$$+ 2(\|M\| + \mu)^2/\mu(F_{mon}(\boldsymbol{\omega}) - F_{mon}(\boldsymbol{\omega}'))^\top(\boldsymbol{\omega} - \boldsymbol{\omega}') \tag{19}$$

$$= O(L)(F_grad(\boldsymbol{\omega}) - F_grad(\boldsymbol{\omega}'))^\top(\boldsymbol{\omega} - \boldsymbol{\omega}') \tag{20}$$

$$+ O(L)(F_{mon}(\boldsymbol{\omega}) - F_{mon}(\boldsymbol{\omega}'))^\top(\boldsymbol{\omega} - \boldsymbol{\omega}') \tag{21}$$

$$= O(L)(F(\boldsymbol{\omega}) - F(\boldsymbol{\omega}'))^\top(\boldsymbol{\omega} - \boldsymbol{\omega}') \tag{22}$$

where for the second inequality we used that a $(L + \mu)$-Lipschitz convex function is $(L + \mu)$-cocoercive and Proposition 2. $\square$

## C   Proof of Theorems

### C.1   Proof of Theorem 1

*Proof.* We consider the following stochastic optimization problem,

$$\frac{1}{n}\sum_{i=1}^{n}\frac{\epsilon}{2}\theta_i^2 + \boldsymbol{\theta}^\top A_i\boldsymbol{\varphi} - \frac{\epsilon}{2}\varphi_i^2 = \frac{1}{n}\sum_{i=1}^{n}\frac{\epsilon}{2}\|A_i\boldsymbol{\theta}\|^2 + \boldsymbol{\theta}^\top A_i\boldsymbol{\varphi} - \frac{\epsilon}{2}\|A_i\boldsymbol{\varphi}\|^2 \tag{23}$$

where $[A_i]_{kl} = 1$ if $k = l = i$ and 0 otherwise. Note that $(A_i)^\top = A_i$ for $1 \leq i \leq n$. Let us consider the extragradient method where to compute an unbiased estimator of the gradients at $(\boldsymbol{\theta}, \boldsymbol{\varphi})$ we sample $i \in \{1, \ldots, n\}$ and use $[A_i\boldsymbol{\theta}, A_i\boldsymbol{\varphi}]$ as estimator of the vector flow.

In this proof we note, $A_I := \sum_{i\in I} A_i$ and $\boldsymbol{\theta}^{(I)}$ the vector such that $[\boldsymbol{\theta}^{(I)}]_i = [\boldsymbol{\theta}]_i$ if $i \in I$ and 0 otherwise. Note that $A_I\boldsymbol{\theta} = \boldsymbol{\theta}^{(I)}$ and that $A_I A_J = A_{I\cap J}$.

Thus the extragradient update rule can be noted as

$$\begin{cases} \boldsymbol{\theta}_{t+1} = (1 - \eta A_I\epsilon)\boldsymbol{\theta}_t - \eta A_I((1 - \eta A_J\epsilon)\boldsymbol{\varphi}_t + \eta A_J\boldsymbol{\theta}_t) \\ \boldsymbol{\varphi}_{t+1} = (1 - \eta A_I\epsilon)\boldsymbol{\varphi}_t + \eta A_I((1 - \eta A_J\epsilon)\boldsymbol{\theta}_t - \eta A_J\boldsymbol{\varphi}_t) \end{cases} \tag{24}$$

where $I$ is the mini-batch sampled (without replacement) for the update and $J$ the mini-batch sampled (without replacement) for the extrapolation.

We can thus notice that, when $I \cap J = \emptyset$, we have

$$\begin{cases} \boldsymbol{\theta}_{t+1} = \boldsymbol{\theta}_t - \eta\epsilon\boldsymbol{\theta}_t^{(I)} - \eta\boldsymbol{\varphi}_t^{(I)} \\ \boldsymbol{\varphi}_{t+1} = \boldsymbol{\varphi}_t - \eta\epsilon\boldsymbol{\varphi}_t^{(I)} + \eta\boldsymbol{\theta}_t^{(I)} \,, \end{cases} \tag{25}$$

and otherwise,

$$\begin{cases} \boldsymbol{\theta}_{t+1} = \boldsymbol{\theta}_t - \eta\epsilon\boldsymbol{\theta}_t^{(I)} - \eta\boldsymbol{\varphi}_t^{(I)} - \eta^2(\boldsymbol{\theta}_t^{(I\cap J)} - \epsilon\boldsymbol{\varphi}_t^{(I\cap J)}) \\ \boldsymbol{\varphi}_{t+1} = \boldsymbol{\varphi}_t - \eta\epsilon\boldsymbol{\varphi}_t^{(I)} + \eta\boldsymbol{\theta}_t^{(I)} - \eta^2(\boldsymbol{\varphi}_t^{(I\cap J)} + \epsilon\boldsymbol{\theta}_t^{(I\cap J)}) \,. \end{cases} \tag{26}$$

The intuition is that, on one hand, when $I \cap J = \emptyset$ (which happens with high probability when $|I| << n$, e.g., when $|I| = 1$, $\mathbb{P}(I \cap J = \emptyset) = 1 - 1/n$), the algorithm performs an update that get away from the Nash equilibrium when $2\epsilon \geq \eta$:

$$(25) \Rightarrow N_{t+1} = N_t + (\eta^2 \epsilon^2 + \eta^2 - 2\eta\epsilon) N_t^{(I)}, \tag{27}$$

where $N_t := \|\boldsymbol{\theta}_t\|^2 + \|\boldsymbol{\varphi}_t\|^2$ and $N_t^{(I)} := \|\boldsymbol{\theta}_t^{(I)}\|^2 + \|\boldsymbol{\varphi}_t^{(I)}\|^2$. On the other hand, The updates that provide improvement only happen when $I \cap J$ is large (which happen with low probability, e.g., when $|I| = 1$, $\mathbb{P}(I \cap J \neq \emptyset) = 1/n$):

$$(26) \Rightarrow N_{t+1} = N_t - N_t^{(I)}(2\eta\epsilon - \eta^2(1+\epsilon^2)) - N_t^{(I \cap J)}(2\eta^2 - \eta^4(1+\epsilon^2)) \tag{28}$$

Conditioning on $\boldsymbol{\theta}_t$ and $\boldsymbol{\varphi}_t$, we get that

$$\mathbb{E}[N_t^{(I \cap J)}|\boldsymbol{\theta}_t, \boldsymbol{\varphi}_t] = \sum_{i=1}^{n} \mathbb{P}(i \in I \cap J)([\boldsymbol{\theta}_t]_i^2 + [\boldsymbol{\varphi}_t]_i^2) \quad \text{and} \quad \mathbb{P}(i \in I \cap J) = \mathbb{P}(i \in I)\mathbb{P}(i \in J) = \frac{|I|^2}{n^2}. \tag{29}$$

Leading to,

$$\mathbb{E}[N_t^{(I \cap J)}|\boldsymbol{\theta}_t, \boldsymbol{\varphi}_t] = \frac{|I|^2}{n^2} \sum_{i=1}^{n} ([\boldsymbol{\theta}_t]_i^2 + [\boldsymbol{\varphi}_t]_i^2) = \frac{|I|^2}{n^2} N_t \quad \text{and} \quad \mathbb{E}[N_t^{(I)}|\boldsymbol{\theta}_t, \boldsymbol{\varphi}_t] = \frac{|I|}{n} N_t. \tag{30}$$

Plugging these expectations in (28), we get that,

$$\mathbb{E}[N_{t+1}] = \left(1 - \tfrac{|I|}{n}(2\eta\epsilon - \eta^2(1+\epsilon^2)) - \tfrac{|I|^2}{n^2}(2\eta^2 - \eta^4(1+\epsilon^2))\right) \mathbb{E}[N_t]. \tag{31}$$

Consequently for $\eta < \epsilon$ we get,

$$\mathbb{E}[N_{t+1}] \geq \left(1 - 2\eta^2 \frac{|I|^2}{n^2} + \eta^2 \frac{|I|}{n}\right) \mathbb{E}[N_t]. \tag{32}$$

To sum-up, if $|I|$ is not large enough (more precisely if $2|I| \leq n$), we have the geometric divergence of the quantity $\mathbb{E}[N_t] := \mathbb{E}[\|\boldsymbol{\theta}_t\|^2 + \|\boldsymbol{\varphi}_t\|^2]$ for any $\eta \geq \epsilon$. $\qquad \square$

## C.2 Proof of Theorem 2

**Setting of the Proof.** We will prove a slightly more general result than Theorem 2. We will work in the context of monotone operator. Let us consider the *general* extrapolation update rule,

$$\begin{cases} \text{Extrapolation:} & \boldsymbol{\omega}_{t+\frac{1}{2}} = \boldsymbol{\omega}_t - \eta_t \boldsymbol{g}_t \\ \qquad \text{Update:} & \boldsymbol{\omega}_{t+1} = \boldsymbol{\omega}_t - \eta_t \boldsymbol{g}_{t+1/2}, \end{cases} \tag{33}$$

where $\boldsymbol{g}_t$ depends on $\boldsymbol{\omega}_t$ and $\boldsymbol{g}_{t+1/2}$ depends on $\boldsymbol{\omega}_{t+1/2}$. For instance, $\boldsymbol{g}_t$ can either be $F(\boldsymbol{\omega}_t)$, $F_{i_t}(\boldsymbol{\omega}_t)$ or the SVRG estimate defined in (45).

This update rule generalizes (EG) for 2-player games (2P-G) and ExtraSVRG (Alg. 2).

Let us first state a lemma standard in convex analysis (see for instance (Boyd and Vandenberghe, 2004)),

**Lemma 1.** *Let $\boldsymbol{\omega} \in \Omega$ and $\boldsymbol{\omega}^+ := P_\Omega(\boldsymbol{\omega} + \boldsymbol{u})$ then for all $\boldsymbol{\omega}' \in \Omega$ we have,*

$$\|\boldsymbol{\omega}^+ - \boldsymbol{\omega}'\|_2^2 \leq \|\boldsymbol{\omega} - \boldsymbol{\omega}'\|_2^2 + 2\boldsymbol{u}^\top(\boldsymbol{\omega}^+ - \boldsymbol{\omega}') - \|\boldsymbol{\omega}^+ - \boldsymbol{\omega}\|_2^2. \tag{34}$$

***Proof of Lemma 1.*** We start by simply developing,

$$\|\boldsymbol{\omega}^+ - \boldsymbol{\omega}'\|_2^2 = \|(\boldsymbol{\omega}^+ - \boldsymbol{\omega}) + (\boldsymbol{\omega} - \boldsymbol{\omega}')\|_2^2 = \|\boldsymbol{\omega} - \boldsymbol{\omega}'\|_2^2 + 2(\boldsymbol{\omega}^+ - \boldsymbol{\omega})^\top(\boldsymbol{\omega} - \boldsymbol{\omega}') + \|\boldsymbol{\omega}^+ - \boldsymbol{\omega}\|_2^2$$
$$= \|\boldsymbol{\omega} - \boldsymbol{\omega}'\|_2^2 + 2(\boldsymbol{\omega}^+ - \boldsymbol{\omega})^\top(\boldsymbol{\omega}^+ - \boldsymbol{\omega}') - \|\boldsymbol{\omega}^+ - \boldsymbol{\omega}\|_2^2\,.$$

Then since $\boldsymbol{\omega}^+$ is the projection onto the convex set $\Omega$ of $\boldsymbol{\omega} + \boldsymbol{u}$ we have that $(\boldsymbol{\omega}^+ - (\boldsymbol{\omega} + \boldsymbol{u}))^\top(\boldsymbol{\omega}^+ - \boldsymbol{\omega}') \leq 0\,, \ \forall\, \boldsymbol{\omega}' \in \Omega$, leading to the result of the Lemma. $\qquad\square$

**Lemma 2.** *If $F$ is $(\mu_{\boldsymbol{\theta}}, \mu_{\boldsymbol{\varphi}})$-strongly monotone for any $\boldsymbol{\omega}, \boldsymbol{\omega}', \boldsymbol{\omega}'' \in \Omega$ we have,*

$$\mu_{\boldsymbol{\theta}}\left(\|\boldsymbol{\theta} - \boldsymbol{\theta}''\|_2^2 - 2\|\boldsymbol{\theta}' - \boldsymbol{\theta}\|_2^2\right) + \mu_{\boldsymbol{\varphi}}\left(\|\boldsymbol{\varphi} - \boldsymbol{\varphi}''\|_2^2 - 2\|\boldsymbol{\varphi}' - \boldsymbol{\varphi}\|_2^2\right) \leq 2(F(\boldsymbol{\omega}') - F(\boldsymbol{\omega}''))^\top(\boldsymbol{\omega}' - \boldsymbol{\omega}'')\,, \quad (35)$$

*where we noted $\boldsymbol{\omega} := (\boldsymbol{\theta}, \boldsymbol{\varphi})$.*

*Proof.* By $(\mu_{\boldsymbol{\theta}}, \mu_{\boldsymbol{\varphi}})$-strong monotonicity,

$$2\mu_{\boldsymbol{\theta}}\|\boldsymbol{\theta}' - \boldsymbol{\theta}''\|_2^2 + 2\mu_{\boldsymbol{\varphi}}\|\boldsymbol{\varphi}' - \boldsymbol{\varphi}''\|_2^2 \leq 2(F(\boldsymbol{\omega}'') - F(\boldsymbol{\omega}''))^\top(\boldsymbol{\omega}' - \boldsymbol{\omega}'') \qquad (36)$$

and then we use the inequality $2\|\boldsymbol{a}' - \boldsymbol{a}''\|_2^2 \geq \|\boldsymbol{a} - \boldsymbol{a}''\|_2^2 - 2\|\boldsymbol{a}' - \boldsymbol{a}\|_2^2$ to get the result claimed. $\qquad\square$

Using this update rule we can thus deduce the following lemma, the derivation of this lemma is very similar from the derivation of Harker and Pang (1990, Lemma 12.1.10).

**Lemma 3.** *Considering the update rule (33), we have for any $\boldsymbol{\omega} \in \Omega$ and any $t \geq 0$,*

$$2\eta_t \boldsymbol{g}_{t+1/2}^\top(\boldsymbol{\omega}_{t+1/2} - \boldsymbol{\omega}) \leq \|\boldsymbol{\omega}_t - \boldsymbol{\omega}\|_2^2 - \|\boldsymbol{\omega}_{t+1} - \boldsymbol{\omega}\|_2^2 - \|\boldsymbol{\omega}_{t+1/2} - \boldsymbol{\omega}_t\|_2^2 + \eta_t^2\|\boldsymbol{g}_t - \boldsymbol{g}_{t+1/2}\|_2^2\,. \quad (37)$$

*Proof.* By applying Lem. 1 for $(\boldsymbol{\omega}, \boldsymbol{u}, \boldsymbol{\omega}^+, \boldsymbol{\omega}') = (\boldsymbol{\omega}_t, -\eta_t \boldsymbol{g}_{t+1/2}, \boldsymbol{\omega}_{t+1}, \boldsymbol{\omega})$ and $(\boldsymbol{\omega}, \boldsymbol{u}, \boldsymbol{\omega}^+, \boldsymbol{\omega}') = (\boldsymbol{\omega}_t, -\eta_t \boldsymbol{g}_t, \boldsymbol{\omega}_{t+1/2}, \boldsymbol{\omega}_{t+1})$, we get,

$$\|\boldsymbol{\omega}_{t+1} - \boldsymbol{\omega}\|_2^2 \leq \|\boldsymbol{\omega}_t - \boldsymbol{\omega}\|_2^2 - 2\eta_t \boldsymbol{g}_{t+1/2}^\top(\boldsymbol{\omega}_{t+1} - \boldsymbol{\omega}) - \|\boldsymbol{\omega}_{t+1} - \boldsymbol{\omega}_t\|_2^2\,, \qquad (38)$$

and

$$\|\boldsymbol{\omega}_{t+1/2} - \boldsymbol{\omega}_{t+1}\|_2^2 \leq \|\boldsymbol{\omega}_t - \boldsymbol{\omega}_{t+1}\|_2^2 - 2\eta_t \boldsymbol{g}_t^\top(\boldsymbol{\omega}_{t+1/2} - \boldsymbol{\omega}_{t+1}) - \|\boldsymbol{\omega}_{t+1/2} - \boldsymbol{\omega}_t\|_2^2\,. \qquad (39)$$

Summing (38) and (39) we get,

$$\|\boldsymbol{\omega}_{t+1} - \boldsymbol{\omega}\|_2^2 \leq \|\boldsymbol{\omega}_t - \boldsymbol{\omega}\|_2^2 - 2\eta_t \boldsymbol{g}_{t+1/2}^\top(\boldsymbol{\omega}_{t+1} - \boldsymbol{\omega}) \qquad (40)$$
$$- 2\eta_t \boldsymbol{g}_t^\top(\boldsymbol{\omega}_{t+1/2} - \boldsymbol{\omega}_{t+1}) - \|\boldsymbol{\omega}_{t+1/2} - \boldsymbol{\omega}_t\|_2^2 - \|\boldsymbol{\omega}_{t+1/2} - \boldsymbol{\omega}_{t+1}\|_2^2 \qquad (41)$$
$$= \|\boldsymbol{\omega}_t - \boldsymbol{\omega}\|_2^2 - 2\eta_t \boldsymbol{g}_{t+1/2}^\top(\boldsymbol{\omega}_{t+1/2} - \boldsymbol{\omega}) - \|\boldsymbol{\omega}_{t+1/2} - \boldsymbol{\omega}_t\|_2^2 - \|\boldsymbol{\omega}_{t+1/2} - \boldsymbol{\omega}_{t+1}\|_2^2$$
$$- 2\eta_t(\boldsymbol{g}_t - \boldsymbol{g}_{t+1/2})^\top(\boldsymbol{\omega}_{t+1/2} - \boldsymbol{\omega}_{t+1})\,. \qquad (42)$$

Then, we can use Young's inequality $-2a^\top b \leq \|a\|_2^2 + \|b\|_2^2$ to get,

$$\|\boldsymbol{\omega}_{t+1} - \boldsymbol{\omega}\|_2^2 \leq \|\boldsymbol{\omega}_t - \boldsymbol{\omega}\|_2^2 - 2\eta_t \boldsymbol{g}_{t+1/2}^\top(\boldsymbol{\omega}_{t+1/2} - \boldsymbol{\omega}) + \eta_t^2\|\boldsymbol{g}_t - \boldsymbol{g}_{t+1/2}\|_2^2$$
$$+ \|\boldsymbol{\omega}_{t+1/2} - \boldsymbol{\omega}_{t+1}\|_2^2 - \|\boldsymbol{\omega}_{t+1/2} - \boldsymbol{\omega}_t\|_2^2 - \|\boldsymbol{\omega}_{t+1/2} - \boldsymbol{\omega}_{t+1}\|_2^2 \qquad (43)$$
$$= \|\boldsymbol{\omega}_t - \boldsymbol{\omega}\|_2^2 - 2\eta_t \boldsymbol{g}_{t+1/2}^\top(\boldsymbol{\omega}_{t+1/2} - \boldsymbol{\omega}) + \eta_t^2\|\boldsymbol{g}_t - \boldsymbol{g}_{t+1/2}\|_2^2 - \|\boldsymbol{\omega}_{t+1/2} - \boldsymbol{\omega}_t\|_2^2\,. \qquad (44)$$

$\qquad\square$

Note that if we would have set $\boldsymbol{g}_t = \boldsymbol{0}$ and $\boldsymbol{g}_{t+1/2}$ any estimate of the gradient at $\boldsymbol{\omega}_t$ we recover the standard lemma for gradient method.

Let us consider *unbiased* estimates of the gradient,

$$\boldsymbol{g}_i(\boldsymbol{\omega}) := \frac{1}{n\pi_i}\left(F_i(\boldsymbol{\omega}) - \boldsymbol{\alpha}_i\right) + \bar{\boldsymbol{\alpha}}, \tag{45}$$

where $\bar{\boldsymbol{\alpha}} := \frac{1}{n}\sum_{j=1}^{n}\boldsymbol{\alpha}_j$, the index $i$ are (potentially) non-uniformly sampled from $\{1, \ldots, n\}$ with replacement according to $\boldsymbol{\pi}$ and $F(\boldsymbol{\omega}) := \frac{1}{n}\sum_{j=1}^{n}F_j(\boldsymbol{\omega})$. Hence we have that $\mathbb{E}[\boldsymbol{g}_i(\boldsymbol{\omega})] = F(\boldsymbol{\omega})$, where the expectation is taken with respect to the index $i$ sampled from $\boldsymbol{\pi}$.

We will consider a class of algorithm called *uniform memorization algorithms* first introduced by (Hofmann et al., 2015). This class of algorithms describes a large subset of variance reduced algorithms taking advantage of the finite sum formulation such as SAGA (Defazio et al., 2014), SVRG (Johnson and Zhang, 2013) or $q$-SAGA and $\mathcal{N}$-SAGA (Hofmann et al., 2015). In this work, we will use a slightly more general definition of such algorithm in order to be able to handle extrapolation steps:

**Definition 9** (Extension of (Hofmann et al., 2015)). *A uniform $q$-memorization algorithm evolves iterates $(\boldsymbol{\omega}_t)$ according to (33), with $\boldsymbol{g}_t$ defined in (45) and selecting in each iteration $t$ a random index set $J_t$ of memory locations to update according to,*

$$\boldsymbol{\alpha}_k^{(0)} := F_k(\boldsymbol{\omega}_0), \quad \boldsymbol{\alpha}_k^{(t+1/2)} := \boldsymbol{\alpha}_k^{(t)}, \ \forall k \in \{1, \ldots, n\} \quad and \quad \boldsymbol{\alpha}_k^{(t+1)} := \begin{cases} F_k(\boldsymbol{\omega}_t) & if \ k \in J_t \\ \boldsymbol{\alpha}_k^{(t)} & otherwise. \end{cases} \tag{46}$$

*such that any $k$ has the same probability $q/n$ to be updated, i.e., $P\{k\} = \sum_{J_t, k\in J_t} P(J_t) = q/n$, $\forall k \in \{1, \ldots, n\}$.*

In the case of SVRG, either $J_t = \emptyset$ or $J_t = \{1, \ldots, n\}$ (when we update the snapshot).

We have the following lemmas,

**Lemma 4.** *For any $t \geq 0$, if we consider a $q$-memorization algorithm we have*

$$\mathbb{E}[\|\boldsymbol{g}_t - \boldsymbol{g}_{t+1/2}\|^2] \leq 10\mathbb{E}[\|\frac{1}{n\pi_i}(F_i(\boldsymbol{\omega}^*) - \boldsymbol{\alpha}_i^{(t)})\|^2] + 10\mathbb{E}[\|\frac{1}{n\pi_i}(F_i(\boldsymbol{\omega}^*) - F_i(\boldsymbol{\omega}_t))\|^2] + 5\bar{L}^2\mathbb{E}[\|\boldsymbol{\omega}_t - \boldsymbol{\omega}_{t+1/2}\|^2].$$

*Proof.* We use an extended version of Young's inequality: $\|\sum_{i=1}^{k}\boldsymbol{a}_i\|^2 \leq k\sum_{i=1}^{k}\|\boldsymbol{a}_i\|^2$,

$$\|\sum_{i=1}^{k}\boldsymbol{a}_i\|^2 = \sum_{i,j=1}^{k}\boldsymbol{a}_i^\top\boldsymbol{a}_j$$

$$\leq \frac{1}{2}\sum_{i,j=1}^{k}\|\boldsymbol{a}_i\|^2 + \|\boldsymbol{a}_j\|^2$$

$$= k\sum_{i=1}^{k}\|\boldsymbol{a}_i\|^2,$$

where we used that $2\boldsymbol{a}^\top\boldsymbol{b} \leq +\|\boldsymbol{a}\|^2 + \|\boldsymbol{b}\|^2$. We combine Young's inequality with the definition of $q$-memorization algorithm: $\boldsymbol{g}_t = \frac{1}{n\pi_i}(F_i(\boldsymbol{\omega}_t) - \bar{\boldsymbol{\alpha}}_i^{(t)})$ and $\boldsymbol{g}_{t+1/2} = \frac{1}{n\pi_j}(F_j(\boldsymbol{\omega}_{t+1/2}) - \bar{\boldsymbol{\alpha}}_j^{(t)})$ to get (we omit

the $t$ subscript for $i$ and $j$ and we note $\bar{\boldsymbol{\alpha}}_i^{(t)} := \boldsymbol{\alpha}_i^{(t)} - n\pi_i \bar{\boldsymbol{\alpha}}^{(t)}$),

$$
\begin{aligned}
\|\boldsymbol{g}_t - \boldsymbol{g}_{t+1/2}\|^2 &= \|\tfrac{1}{n\pi_i}(F_i(\boldsymbol{\omega}_t) - \bar{\boldsymbol{\alpha}}_i^{(t)}) - \tfrac{1}{n\pi_j}(F_j(\boldsymbol{\omega}_{t+1/2}) - \bar{\boldsymbol{\alpha}}_j^{(t)})\|^2 \\
&= \|\tfrac{1}{n\pi_i}(F_i(\boldsymbol{\omega}_t) - \bar{\boldsymbol{\alpha}}_i^{(t)}) + \tfrac{1}{n\pi_j}(F_j(\boldsymbol{\omega}_t) - F_j(\boldsymbol{\omega}_{t+1/2})) + \tfrac{1}{n\pi_j}(\bar{\boldsymbol{\alpha}}_j^{(t)} - F_j(\boldsymbol{\omega}_t))\|^2 \\
&\leq 5\mathbb{E}[\|\tfrac{1}{n\pi_i}(F_i(\boldsymbol{\omega}^*) - \bar{\boldsymbol{\alpha}}_i^{(t)}))\|^2] + 5\mathbb{E}[\|\tfrac{1}{n\pi_j}(\bar{\boldsymbol{\alpha}}_j^{(t)} - F_j(\boldsymbol{\omega}^*))\|^2] \\
&\quad + 5\mathbb{E}[\|\tfrac{1}{n\pi_i}(F_i(\boldsymbol{\omega}^*) - F_i(\boldsymbol{\omega}_t))\|^2] + 5\mathbb{E}[\|\tfrac{1}{n\pi_j}(F_j(\boldsymbol{\omega}^*) - F_j(\boldsymbol{\omega}_t))\|^2] \\
&\quad + 5\mathbb{E}[\|\tfrac{1}{n\pi_j}(F_j(\boldsymbol{\omega}_t) - F_j(\boldsymbol{\omega}_{t+1/2}))\|^2]
\end{aligned}
$$

Notice that since $i_t$ and $j_t$ are independently sampled from the same distribution we have

$$
\mathbb{E}[\tfrac{1}{n^2\pi_{j_t}^2}\|F_{j_t}(\boldsymbol{\omega}^*) - \boldsymbol{\alpha}_{j_t}^{(t)}\|^2] = \mathbb{E}[\tfrac{1}{n^2\pi_{i_t}^2}\|F_{i_t}(\boldsymbol{\omega}^*) - \boldsymbol{\alpha}_{i_t}^{(t)}\|^2]. \tag{47}
$$

Note that we have (using that $\mathbb{E}[F_i(\boldsymbol{\omega}^*)] = 0$ and $\mathbb{E}[\boldsymbol{\alpha}_i^{(t)}] = \bar{\boldsymbol{\alpha}}^{(t)}$),

$$
\mathbb{E}[\|\tfrac{1}{n\pi_i}(F_i(\boldsymbol{\omega}^*) - \bar{\boldsymbol{\alpha}}_i^{(t)})\|^2] = \mathbb{E}[\|\tfrac{1}{n\pi_i}(F_i(\boldsymbol{\omega}^*) - \boldsymbol{\alpha}_i^{(t)})\|^2] - \|\bar{\boldsymbol{\alpha}}^{(t)}\|^2 \leq \mathbb{E}[\|\tfrac{1}{n\pi_i}(F_i(\boldsymbol{\omega}^*) - \boldsymbol{\alpha}_i^{(t)})\|^2] \tag{48}
$$

By assuming that each $F_i$ is $L_i$-Lipschitz we get,

$$
\mathbb{E}[\tfrac{1}{n^2\pi_{j_t}^2}\|F_j(\boldsymbol{\omega}_t) - F_j(\boldsymbol{\omega}_{t+1/2})\|^2] = \frac{1}{n^2}\sum_{j=1}^n \frac{1}{\pi_j}\mathbb{E}[\|F_j(\boldsymbol{\omega}_t) - F_j(\boldsymbol{\omega}_{t+1/2})\|^2] \tag{49}
$$

$$
\leq \frac{1}{n^2}\sum_{j=1}^n \frac{L_j^2}{\pi_j}\mathbb{E}[\|\boldsymbol{\omega}_t - \boldsymbol{\omega}_{t+1/2}\|^2] \tag{50}
$$

$$
= \bar{L}^2\mathbb{E}[\|\boldsymbol{\omega}_t - \boldsymbol{\omega}_{t+1/2}\|^2], \tag{51}
$$

where $\bar{L}^2 := \frac{1}{n^2}\sum_{i=1}^n \frac{L_i^2}{\pi_j}$. Note that $\boldsymbol{\omega}_t$ and $\boldsymbol{\omega}_{t+1/2}$ do not depend on $j_t$ (which is the index sampled for the update step), that is not the case for $i$ (the index for the extrapolation step) since $\boldsymbol{\omega}_{t+1/2}$ is the result of the extrapolation. $\qquad\square$

This lemma make appear the quantity $\mathbb{E}[\|\tfrac{1}{n\pi_i}(F_i(\boldsymbol{\omega}^*) - \bar{\boldsymbol{\alpha}}_i^{(t)})\|^2]$ that we need to bound. In order to do that we prove the following lemma,

**Lemma 5.** *Let $(\boldsymbol{\alpha}_j^{(t)})$ be updated according to the rules of a q-uniform memorization algorithm (Def. 9). Let us note $H_t := \frac{1}{n}\sum_{i=1}^n \frac{1}{n\pi_i}\|F_i(\boldsymbol{\omega}^*) - \boldsymbol{\alpha}_i^{(t)}\|^2$. For any $t \in \mathbb{N}$,*

$$
\mathbb{E}[H_{t+1}] = \frac{q}{n}\mathbb{E}[\|\tfrac{1}{n\pi_{i_t}}(F_{i_t}(\boldsymbol{\omega}_t) - F_{i_t}(\boldsymbol{\omega}^*))\|^2] + \frac{n-q}{n}\mathbb{E}[H_t]. \tag{52}
$$

*Proof.* We will use the definition of $q$-uniform memorization algorithms (saying that $\boldsymbol{\alpha}_i$ is updated at time $t+1$ with probability $q/n$). We call this event "$i$ updated",

$$
\mathbb{E}[H_{t+1}] := \mathbb{E}[\frac{1}{n} \sum_{i=1}^{n} \frac{1}{n\pi_i} \|\boldsymbol{\alpha}_i^{(t+1)} - F_i(\boldsymbol{\omega}^*)\|^2]
$$

$$
= \frac{1}{n}\mathbb{E}[\sum_{i \text{ updated}} \frac{1}{n\pi_i} \|\boldsymbol{\alpha}_i^{(t+1)} - F_i(\boldsymbol{\omega}^*)\|^2 + \sum_{i \text{ not updated}} \frac{1}{n\pi_i} \|\boldsymbol{\alpha}_i^{(t+1)} - F_i(\boldsymbol{\omega}^*)\|^2]
$$

$$
= \frac{1}{n}\mathbb{E}[\sum_{i \text{ updated}} \frac{1}{n\pi_i} \|F_i(\boldsymbol{\omega}_t) - F_i(\boldsymbol{\omega}^*)\|^2 + \sum_{i \text{ not updated}} \frac{1}{n\pi_i} \|\boldsymbol{\alpha}_i^{(t)} - F_i(\boldsymbol{\omega}^*)\|^2]
$$

$$
= \frac{1}{n} \sum_{i=1}^{n} \mathbf{P}(i \text{ updated})\frac{1}{n\pi_i}\mathbb{E}[\|F_i(\boldsymbol{\omega}_t) - F_i(\boldsymbol{\omega}^*)\|^2 + \frac{1}{n} \sum_{i=1}^{n} \mathbf{P}(i \text{ not updated})\frac{1}{n\pi_i}\mathbb{E}[\|\boldsymbol{\alpha}_i^{(t)} - F_i(\boldsymbol{\omega}^*)\|^2]
$$

$$
= \frac{q}{n}\mathbb{E}[\|\frac{1}{n\pi_{i_t}}(F_{i_t}(\boldsymbol{\omega}_t) - F_{i_t}(\boldsymbol{\omega}^*))\|^2] + \frac{n-q}{n}\mathbb{E}[H_t]
$$

$\square$

Using all these lemmas we can prove our theorem.

**Theorem' 2.** *Under Assumption 1, after $t$ iterations, the iterate $\boldsymbol{\omega}_t$ computed by a $q$-memorization algorithm with step-sizes $(\eta_\theta, \eta_\phi) \leq \left((40\bar{\ell}_{\boldsymbol{\theta}})^{-1}, (40\bar{\ell}_{\boldsymbol{\varphi}})^{-1}\right)$ verifies:*

$$
\mathbb{E}[\|\boldsymbol{\omega}_t - \boldsymbol{\omega}^*\|_2^2] \leq \left(1 - \min\left\{\frac{\eta\mu}{2} + \frac{9\eta^2\gamma^2}{10}, \frac{2q}{5n}\right\}\right)^t \mathbb{E}[\|\boldsymbol{\omega}_0 - \boldsymbol{\omega}^*\|_2^2]. \tag{53}
$$

*Proof.* In this proof we will consider a constant step-size $\eta_t = (\eta_\theta, \eta_\phi)$. For simplicity of notations we will consider the notation,

$$
\bar{L}^2\|\boldsymbol{\omega}\|^2 := \bar{L}_{\boldsymbol{\theta}}^2\|\boldsymbol{\theta}\|^2 + \bar{L}_{\boldsymbol{\varphi}}^2\|\boldsymbol{\varphi}\|^2, \quad \eta^2\|\boldsymbol{\omega}\|^2 := \eta_{\boldsymbol{\theta}}^2\|\boldsymbol{\theta}\|^2 + \eta_{\boldsymbol{\varphi}}^2\|\boldsymbol{\varphi}\|^2, \quad \mu\|\boldsymbol{\omega}\|^2 := \mu_{\boldsymbol{\theta}}^2\|\boldsymbol{\theta}\|^2 + \mu_{\boldsymbol{\varphi}}^2\|\boldsymbol{\varphi}\|^2
$$

$$
\eta\mu = (\eta_{\boldsymbol{\theta}}\mu_{\boldsymbol{\theta}}, \eta_{\boldsymbol{\varphi}}\mu_{\boldsymbol{\varphi}}), \quad \sigma\bar{L}^2 = (\sigma_{\boldsymbol{\theta}}\bar{L}_{\boldsymbol{\theta}}^2, \sigma_{\boldsymbol{\varphi}}\bar{L}_{\boldsymbol{\varphi}}^2) \quad \text{and} \quad \eta^2\bar{L}^2 = (\eta_{\boldsymbol{\theta}}^2\bar{L}_{\boldsymbol{\theta}}^2, \eta_{\boldsymbol{\varphi}}^2\bar{L}_{\boldsymbol{\varphi}}^2).
$$

We start by recalling Lemma 3,

$$
\|\boldsymbol{\omega}_{t+1} - \boldsymbol{\omega}^*\|_2^2 \leq \|\boldsymbol{\omega}_t - \boldsymbol{\omega}^*\|_2^2 - 2\eta \boldsymbol{g}_{t+1/2}^{\top}(\boldsymbol{\omega}_{t+1/2} - \boldsymbol{\omega}^*) - (1 - 2\eta\mu)\|\boldsymbol{\omega}_{t+1/2} - \boldsymbol{\omega}_t\|_2^2 + \eta^2\|\boldsymbol{g}_t - \boldsymbol{g}_{t+1/2}\|_2^2. \tag{54}
$$

We can then take the expectation and plug-in the expression of $\mathbb{E}[\|\boldsymbol{g}_t - \boldsymbol{g}_{t+1/2}\|_2^2]$ from Lemma 4,

$$
\mathbb{E}[\|\boldsymbol{\omega}_{t+1} - \boldsymbol{\omega}^*\|_2^2] \leq \mathbb{E}[\|\boldsymbol{\omega}_t - \boldsymbol{\omega}^*\|_2^2] - 2\eta\mathbb{E}[F(\boldsymbol{\omega}_{t+1/2})^{\top}(\boldsymbol{\omega}_{t+1/2} - \boldsymbol{\omega}^*)] - (1 - 2\eta\mu - 5\eta^2\bar{L}^2)]\mathbb{E}[\|\boldsymbol{\omega}_{t+1/2} - \boldsymbol{\omega}_t\|_2^2]
$$

$$
+ \eta^2(10\mathbb{E}[\|\frac{1}{n\pi_i}(F_i(\boldsymbol{\omega}^*) - \boldsymbol{\alpha}_i^{(t)})\|^2] + 10\mathbb{E}[\|\frac{1}{n\pi_i}(F_i(\boldsymbol{\omega}^*) - F_i(\boldsymbol{\omega}_t))\|^2]).
$$

Let us define $\mathcal{L}_t := \mathbb{E}[\|\boldsymbol{\omega}_t - \boldsymbol{\omega}^*\|_2^2] + \sigma\mathbb{E}[H_t]$, where $H_t := \frac{1}{n}\sum_{i=1}^{n}\frac{1}{n\pi_i}\|F_i(\boldsymbol{\omega}^*) - \boldsymbol{\alpha}_i^{(t)}\|^2$. We can combine (54) with Lemma 5 multiplied by a constant $\sigma > 0$ that we will set later to get

$$
\mathcal{L}_{t+1} = \mathbb{E}[\|\boldsymbol{\omega}_{t+1} - \boldsymbol{\omega}^*\|_2^2] + \sigma\mathbb{E}[H_{t+1}]
$$

$$
\leq \mathbb{E}[\|\boldsymbol{\omega}_t - \boldsymbol{\omega}^*\|_2^2] - 2\eta\mathbb{E}[F(\boldsymbol{\omega}_{t+1/2})^{\top}(\boldsymbol{\omega}_{t+1/2} - \boldsymbol{\omega}^*)] - (1 - 2\eta\mu - 5\eta^2\bar{L}^2)\|\boldsymbol{\omega}_{t+1/2} - \boldsymbol{\omega}_t\|_2^2
$$

$$
+ (\frac{\sigma q}{n} + 10\eta^2)\mathbb{E}[\|\frac{1}{n\pi_i}(F_i(\boldsymbol{\omega}^*) - F_i(\boldsymbol{\omega}_t))\|^2 + (\frac{10\eta^2}{\sigma} + \frac{n-q}{n})\sigma\mathbb{E}[H_t].
$$

Since $i_t$ and $j_t$ are independently drawn from the same distribution, we have, $\mathbb{E}[\|\frac{1}{n\pi_i}(F_i(\boldsymbol{\omega}^*) - F_i(\boldsymbol{\omega}_t))\|^2] = \mathbb{E}[\|\frac{1}{n\pi_j}(F_j(\boldsymbol{\omega}^*) - F_j(\boldsymbol{\omega}_t))\|^2$ and thus,

$$
\begin{aligned}
\mathcal{L}_{t+1} &\leq \mathbb{E}[\|\boldsymbol{\omega}_t - \boldsymbol{\omega}^*\|_2^2] - 2\eta\mathbb{E}[F(\boldsymbol{\omega}_{t+1/2})^\top(\boldsymbol{\omega}_{t+1/2} - \boldsymbol{\omega}^*)] - (1 - 2\eta\mu - 5\eta^2\bar{L}^2)\|\boldsymbol{\omega}_{t+1/2} - \boldsymbol{\omega}_t\|_2^2 \\
&\quad + (\tfrac{\sigma q}{n} + 10\eta^2)\mathbb{E}[\|\tfrac{1}{n\pi_j}(F_j(\boldsymbol{\omega}^*) - F_j(\boldsymbol{\omega}_t))\|^2 + (\tfrac{10\eta^2}{\sigma} + \tfrac{n-q}{n})\sigma\mathbb{E}[H_t] \\
&\leq \mathbb{E}[\|\boldsymbol{\omega}_t - \boldsymbol{\omega}^*\|_2^2] - (1 - 2\eta\mu - 5\eta^2\bar{L}^2 - 2(\tfrac{\sigma q}{n} + 10\eta^2)\bar{L}^2)\|\boldsymbol{\omega}_{t+1/2} - \boldsymbol{\omega}_t\|_2^2 \\
&\quad - 2\eta\mathbb{E}[F(\boldsymbol{\omega}_{t+1/2})^\top(\boldsymbol{\omega}_{t+1/2} - \boldsymbol{\omega}^*)] + 2(\tfrac{\sigma q}{n} + 10\eta^2)\mathbb{E}[\|\tfrac{1}{n\pi_j}(F_j(\boldsymbol{\omega}^*) - F_j(\boldsymbol{\omega}_{t+1/2}))\|^2 \\
&\quad + (\tfrac{10\eta^2}{\sigma} + \tfrac{n-q}{n})\sigma\mathbb{E}[H_t] \\
&\leq \mathbb{E}[\|\boldsymbol{\omega}_t - \boldsymbol{\omega}^*\|_2^2] - (1 - 2\eta\mu - 5\eta^2\bar{L}^2 - 2(\tfrac{\sigma q}{n} + 10\eta^2)\bar{L}^2)\|\boldsymbol{\omega}_{t+1/2} - \boldsymbol{\omega}_t\|_2^2 \\
&\quad - 2\eta\mathbb{E}[F(\boldsymbol{\omega}_{t+1/2})^\top(\boldsymbol{\omega}_{t+1/2} - \boldsymbol{\omega}^*)] + (\tfrac{10\eta^2}{\sigma} + \tfrac{n-q}{n})\sigma\mathbb{E}[H_t] \\
&\quad + 2(\tfrac{\sigma q}{n} + 10\eta^2)\mathbb{E}[\tfrac{\ell_j}{n^2\pi_j^2}(F_j(\boldsymbol{\omega}^*) - F_j(\boldsymbol{\omega}_{t+1/2}))^\top(\boldsymbol{\omega}^* - \boldsymbol{\omega}_{t+1/2})]
\end{aligned}
$$

where for the second inequality we used Young's inequality and the Lipchitzness of $F_j$ and for the last one we used the co-coercivity of $F_j$:

$$
\|F_j(\boldsymbol{\omega}) - F_j(\boldsymbol{\omega}')\|^2 \leq \ell_i(F_j(\boldsymbol{\omega}') - F_j(\boldsymbol{\omega}))^\top(\boldsymbol{\omega}' - \boldsymbol{\omega}). \tag{55}
$$

Thus using $\pi_j = \frac{\ell_j}{\sum_j \ell_j}$, we get

$$
\begin{aligned}
\mathcal{L}_{t+1} &\leq \mathbb{E}[\|\boldsymbol{\omega}_t - \boldsymbol{\omega}^*\|_2^2] - (1 - 2\eta\mu - 5\eta^2\bar{L}^2 - 2(\tfrac{\sigma q}{n} + 10\eta^2)\bar{L}^2)\|\boldsymbol{\omega}_{t+1/2} - \boldsymbol{\omega}_t\|_2^2 \\
&\quad - 2\eta\mathbb{E}[F(\boldsymbol{\omega}_{t+1/2})^\top(\boldsymbol{\omega}_{t+1/2} - \boldsymbol{\omega}^*)] + 2(\tfrac{10\eta^2}{\sigma} + \tfrac{n-q}{n})\sigma\mathbb{E}[H_t] \\
&\quad + 2\bar{\ell}(\tfrac{\sigma q}{n} + 10\eta^2)\mathbb{E}[\tfrac{1}{n\pi_j}(F_j(\boldsymbol{\omega}^*) - F_j(\boldsymbol{\omega}_{t+1/2}))^\top(\boldsymbol{\omega}^* - \boldsymbol{\omega}_{t+1/2})] \\
&= \mathbb{E}[\|\boldsymbol{\omega}_t - \boldsymbol{\omega}^*\|_2^2] - (1 - 2\eta\mu - 5\eta^2\bar{L}^2 - 2(\tfrac{\sigma q}{n} + 10\eta^2)\bar{L}^2)\|\boldsymbol{\omega}_{t+1/2} - \boldsymbol{\omega}_t\|_2^2 \\
&\quad - 2\eta\mathbb{E}[F(\boldsymbol{\omega}_{t+1/2})^\top(\boldsymbol{\omega}_{t+1/2} - \boldsymbol{\omega}^*)] + 2\bar{\ell}(\tfrac{\sigma q}{n} + 10\eta^2)\mathbb{E}[F(\boldsymbol{\omega}_{t+1/2})^\top(\boldsymbol{\omega}_{t+1/2} - \boldsymbol{\omega}^*)] \\
&\quad + (\tfrac{10\eta^2}{\sigma} + \tfrac{n-q}{n})\sigma\mathbb{E}[H_t]
\end{aligned}
$$

where $\bar{\ell} := \frac{1}{n}\sum_i \ell_i$. Now we can set $\frac{20\eta^2}{\sigma} = \frac{q}{n}$ to get

$$
\begin{aligned}
\mathcal{L}_{t+1} &\leq \mathbb{E}[\|\boldsymbol{\omega}_t - \boldsymbol{\omega}^*\|_2^2] - (1 - 2\eta\mu - 65\eta^2\bar{L}^2)\|\boldsymbol{\omega}_{t+1/2} - \boldsymbol{\omega}_t\|_2^2 \\
&\quad - \eta(2 - 60\bar{\ell}\eta)\mathbb{E}[F(\boldsymbol{\omega}_{t+1/2})^\top(\boldsymbol{\omega}_{t+1/2} - \boldsymbol{\omega}^*)] + (1 - \tfrac{q}{2n})\sigma\mathbb{E}[H_t].
\end{aligned}
$$

Finally with $\eta \leq \frac{1}{40\bar{\ell}}$ (note that we always have $\bar{\ell} \geq \bar{L}$ because $\ell_i \geq L_i$) we get

$$
\mathcal{L}_{t+1} \leq \mathbb{E}[\|\boldsymbol{\omega}_t - \boldsymbol{\omega}^*\|_2^2] - \eta\frac{1}{2}\mathbb{E}[F(\boldsymbol{\omega}_{t+1/2})^\top(\boldsymbol{\omega}_{t+1/2} - \boldsymbol{\omega}^*)] - \frac{9}{10}\mathbb{E}[\|\boldsymbol{\omega}_{t+1/2} - \boldsymbol{\omega}_t\|_2^2] + (1 - \tfrac{q}{2n})\sigma\mathbb{E}[H_t].
$$

We finaly use the projection-type error bound $\|F_i(\boldsymbol{\omega}_t) - F_i(\boldsymbol{\omega}^*)\|^2 \geq \gamma_i^2\|\boldsymbol{\omega}_t - \boldsymbol{\omega}^*\|^2$ the same way as (Azizian et al., 2019) to get,

$$
\begin{aligned}
\|\boldsymbol{\omega}_{t+1/2} - \boldsymbol{\omega}_t\|_2^2 &= \eta^2\|\tfrac{1}{n\pi_i}(F_i(\boldsymbol{\omega}_t) - \bar{\boldsymbol{\alpha}}_i^{(t)})\|^2 \\
&\geq \frac{\eta^2}{2}\|\tfrac{1}{n\pi_i}(F_i(\boldsymbol{\omega}_t) - F_i(\boldsymbol{\omega}^*))\|^2 - \eta^2\|\tfrac{1}{n\pi_i}(F_i(\boldsymbol{\omega}^*) - \bar{\boldsymbol{\alpha}}_i^{(t)})\|^2 \\
&\geq \frac{\gamma_i^2\eta^2}{2}\|\tfrac{1}{n\pi_i}(\boldsymbol{\omega}_t - \boldsymbol{\omega}^*)\|^2 - \eta^2\|\tfrac{1}{n\pi_i}(F_i(\boldsymbol{\omega}^*) - \bar{\boldsymbol{\alpha}}_i^{(t)})\|^2.
\end{aligned}
$$

Thus we have that,

$$\mathcal{L}_{t+1} \leq \mathbb{E}[\|\boldsymbol{\omega}_t - \boldsymbol{\omega}^*\|_2^2] - \eta\frac{1}{2}\mathbb{E}[F(\boldsymbol{\omega}_{t+1/2})^\top(\boldsymbol{\omega}_{t+1/2} - \boldsymbol{\omega}^*)] - \frac{\bar{\gamma}^2\eta^2}{2}\mathbb{E}[\|\boldsymbol{\omega}_t - \boldsymbol{\omega}^*\|_2^2] + (1 - \frac{q}{2n} + \frac{9q}{100n})\sigma\mathbb{E}[H_t]\,,$$

where $\bar{\gamma}^2 := \frac{1}{n}\sum_{k=1}^n \frac{\gamma_i^2}{n\pi_i}$. We can thus conclude the proof using the strong convexity of $F$,

$$\mathcal{L}_{t+1} \leq \left(1 - \min\left\{\left(\frac{\eta\mu}{2} + \frac{9\eta^2\bar{\gamma}^2}{20}\right), \frac{2q}{5n}\right\}\right)\mathcal{L}_t\,.$$

$\square$

## D  Details on the SVRE–GAN Algorithm

### D.1  Practical Aspect

**Noise dataset.**  Variance reduction is usually performed on finite sum dataset. However, the noise dataset in GANs (sampling from the noise variable $z$ for the generator $G$) is in practice considered as an infinite dataset. We considered several ways to cope with this:

- Infinitely taking new samples from a predefined latent distribution $p_g$. In this case, from a theoretical point of view, in terms of using finite sum formulation, there is no convergence guarantee for SVRE even in the strongly convex case. Moreover, the estimators (63) and (64) are biased estimator of the gradient (as $\boldsymbol{\mu}_D$ and $\boldsymbol{\mu}_G$ do not estimate the full expectation but a finite sum).

- Sampling a different noise dataset at each epoch, i.e. considering a different finite sum at each epoch. In that case, we are performing a variance reduction of this finite sum over the epoch.

- Fix a finite sum noise dataset for the entire training.

In practice, we did not notice any notable difference between the three alternatives.

**Adaptive methods.**  Particular choices such as the optimization method (*e.g.* Adam (Kingma and Ba, 2015)), learning rates, and normalization, have been established in practice as almost *prerequisite* for convergence[7], in contrast to supervised classification problems where they have been shown to only provide a marginal value (Wilson et al., 2017). To our knowledge, SVRE is the only method that works with a constant step size for GANs on non-trivial datasets. This combined with the fact that recent works empirically tune the first moment controlling hyperparameter to 0 ($\beta_1$, see below) and the variance reduction (VR) one ($\beta_2$, see below) to a non-zero value, sheds light on the reason behind the success of Adam on GANs.

However, combining SVRE with adaptive step size scheme on GANs remains an open problem. We first briefly describe the update rule of Adam, and then we propose a new adaptation of it that is more suitable for VR methods, which we refer to as variance reduced Adam (VRAd).

**Adam.**  Adam stores an exponentially decaying average of both past gradients $m_t$ and squared gradients $v_t$, for each parameter of the model:

$$m_t = \beta_1 m_{t-1} + (1 - \beta_1)g_t \tag{56}$$

$$v_t = \beta_2 v_{t-1} + (1 - \beta_2)g_t^2\,, \tag{57}$$

**Algorithm 2** Pseudocode for SVRE-GAN.

---

1: **Input:** dataset $\mathcal{D}$, noise dataset $\mathcal{Z}$ ($|\mathcal{Z}| = |\mathcal{D}| = n$), stopping iteration $T$, learning rates $\eta_D, \eta_G$, generator loss $\mathcal{L}^G$, discriminator loss $\mathcal{L}^D$, mini-batch size B.
2: **Initialize:** $D, G$
3: **for** $e = 0$ **to** $T-1$ **do**
4:     $D^{\mathcal{S}} = D$ and $\boldsymbol{\mu}_D = \frac{1}{n}\sum_{i=1}^{n}\sum_{j=1}^{n}\nabla_D\mathcal{L}^D(G^{\mathcal{S}}, D^{\mathcal{S}}, \mathcal{D}_j, \mathcal{Z}_i)$
5:     $G^{\mathcal{S}} = G$ and $\boldsymbol{\mu}_G = \frac{1}{n}\sum_{i=1}^{n}\nabla_G\mathcal{L}^G(G^{\mathcal{S}}, D^{\mathcal{S}}, \mathcal{Z}_i)$
6:     $N \sim \text{Geom}\left(B/n\right)$                                     (length of the epoch)
7:     **for** $i = 0$ **to** $N-1$ **do**
8:         **Sample** mini-batches $(n_d, n_z)$; do **extrapolation:**
9:         $\tilde{D} = D - \eta_D\boldsymbol{d}_D(G, D, G^{\mathcal{S}}, D^{\mathcal{S}}, n_z)$                          ▷ (63)
10:        $\tilde{G} = G - \eta_G\boldsymbol{d}_G(G, D, G^{\mathcal{S}}, D^{\mathcal{S}}, n_d, n_z)$                      ▷ (64)
11:        **Sample** new mini-batches $(n_d, n_z)$; do **update:**
12:        $D = D - \eta_D\boldsymbol{d}_D(\tilde{G}, \tilde{D}, G^{\mathcal{S}}, D^{\mathcal{S}}, n_z)$                          ▷ (63)
13:        $G = G - \eta_G\boldsymbol{d}_G(\tilde{G}, \tilde{D}, G^{\mathcal{S}}, D^{\mathcal{S}}, n_d, n_z)$                      ▷ (64)
14: **Output:** $G, D$

---

where $\beta_1, \beta_2 \in [0, 1]$, $m_0 = 0$, $v_0 = 0$, and $t = 1, \ldots T$ denotes the iteration. $m_t$ and $v_t$ are respectively the estimates of the first and the second moments of the stochastic gradient. To compensate the bias toward $0$ due to initialization, Kingma and Ba (2015) propose to use bias-corrected estimates of these first two moments:

$$\hat{m}_t = \frac{m_t}{1 - \beta_1^t} \tag{58}$$

$$\hat{v}_t = \frac{v_t}{1 - \beta_2^t}. \tag{59}$$

The Adam update rule can be described as:

$$\boldsymbol{\omega}_{t+1} = \boldsymbol{\omega}_t - \eta\frac{\hat{\boldsymbol{m}}_t}{\sqrt{\hat{\boldsymbol{v}}_t} + \epsilon}. \tag{60}$$

Adam can be understood as an approximate gradient method with a diagonal step size of $\eta_{Adam} := \frac{\eta}{\sqrt{\boldsymbol{v}_t} + \epsilon}$. Since VR methods aim to provide a vanishing $\boldsymbol{v}_t$, they lead to a too large step-size $\eta_{Adam}$ of $\frac{\eta}{\epsilon}$. This could indicate that the update rule of Adam may not be a well-suited method to combine with VR methods.

**VRAd.**   This motivates the introduction of a new Adam-inspired variant of adaptive step sizes that maintain a reasonable size even when $\boldsymbol{v}_t$ vanishes,

$$\boldsymbol{\omega}_{t+1} = \boldsymbol{\omega}_t - \eta\frac{|\hat{\boldsymbol{m}}_t|}{\sqrt{\hat{\boldsymbol{v}}_t} + \epsilon}\hat{\boldsymbol{m}}_t. \tag{VRAd}$$

This adaptive variant of Adam is motivated by the step size $\eta^* = \eta\frac{\boldsymbol{m}_t^2}{\boldsymbol{v}_t}$ derived by Schaul et al. (2013). (VRAd) is simply the square-root of $\eta^*$ in order to stick with Adam's scaling of $\boldsymbol{v}_t$.

## D.2   SVRE-GAN

In order to cope with the issues introduced by the stochastic game formulation of the GAN models, we proposed the SVRE algorithm Alg. 1 which combines SVRG and extragradient method. We refer to

the method of applying SVRE to train GANs as the *SVRE-GAN* method, and we describe it in detail in Alg. 2 (generalizing it with mini-batching, but using uniform probabilities). Assuming that we have $\mathcal{D}[n_d]$ and $\mathcal{Z}[n_z]$, respectively two mini-batches of size $B$ of the true dataset and the noise dataset, we compute $\nabla_D \mathcal{L}^D(G, D, \mathcal{D}[n_d], \mathcal{Z}[n_z])$ and $\nabla_G \mathcal{L}^G(G, D, \mathcal{Z}[n_z])$ the respective mini-batches gradient of the discriminator and the generator:

$$\nabla_D \mathcal{L}^D(G, D, \mathcal{D}[n_d], \mathcal{Z}[n_z]) := \frac{1}{|n_z|} \frac{1}{|n_d|} \sum_{i \in n_z} \sum_{j \in n_d} \nabla_D \mathcal{L}^D(G, D, \mathcal{D}_j, \mathcal{Z}_i) \tag{61}$$

$$\nabla_G \mathcal{L}^G(G, D, \mathcal{Z}[n_z]) := \frac{1}{|n_z|} \sum_{i \in n_z} \nabla_G \mathcal{L}^G(G, D, \mathcal{Z}_i), \tag{62}$$

where $\mathcal{Z}_i$ and $\mathcal{D}_j$ are respectively the $i^{th}$ example of the noise dataset and the $j^{th}$ of the true dataset. Note that $n_z$ and $n_d$ are lists and thus that we allow repetitions in the summations over $n_z$ and $n_d$. The variance reduced gradient of the SVRG method are thus given by:

$$\boldsymbol{d}_D(G, D, G^{\mathcal{S}}, D^{\mathcal{S}}) := \boldsymbol{\mu}_D + \nabla_D \mathcal{L}^D(G, D, \mathcal{D}[n_d], \mathcal{Z}[n_z]) - \nabla_D \mathcal{L}^D(G^{\mathcal{S}}, D^{\mathcal{S}}, \mathcal{D}[n_d], \mathcal{Z}[n_z]) \tag{63}$$

$$\boldsymbol{d}_G(G, D, G^{\mathcal{S}}, D^{\mathcal{S}}) := \boldsymbol{\mu}_G + \nabla_G \mathcal{L}^G(G, D, \mathcal{Z}[n_z]) - \nabla \mathcal{L}^G(G^{\mathcal{S}}, D^{\mathcal{S}}, \mathcal{Z}[n_z]), \tag{64}$$

where $G^{\mathcal{S}}$ and $D^{\mathcal{S}}$ are the snapshots and $\boldsymbol{\mu}_D$ and $\boldsymbol{\mu}_G$ their respective gradients.

Alg. 2 summarizes the SVRG optimization extended to GAN. To obtain that $\mathbb{E}\left[\nabla_{\boldsymbol{\Theta}^{\mathcal{S}}} \mathcal{L}(\boldsymbol{\theta}^{\mathcal{S}}, \boldsymbol{\varphi}^{\mathcal{S}}, \cdot) - \boldsymbol{\mu}\right]$ vanishes, when updating $\boldsymbol{\theta}$ and $\boldsymbol{\varphi}$ where the expectation is over samples of $\mathcal{D}$ and $\mathcal{Z}$ respectively, we use the snapshot networks $\boldsymbol{\theta}^{\mathcal{S}}$ and $\boldsymbol{\varphi}^{\mathcal{S}}$ for the second term in lines $9, 10, 12$ and $13$. Moreover, the noise dataset $\mathcal{Z} \sim p_z$, where $|\mathcal{Z}| = |\mathcal{D}| = n$, is fixed. Empirically we observe that directly sampling from $p_z$ (contrary to fixing the noise dataset and re-sampling it with frequency $m$) does not impact the performance, as $|\mathcal{Z}|$ is usually high.

Note that the double sum in Line 4 can be written as two sums because of the separability of the expectations in typical GAN objectives. Thus the time complexity for calculating $\mu^D$ is still $O(n)$ and not $O(n^2)$ which would be prohibitively expensive.

# E    Restarted SVRE

Alg. 3 describes the restarted version of SVRE presented in § 3.3. With a probability $p$ (fixed) before the computation of $\boldsymbol{\mu}_{\boldsymbol{\varphi}}^{\mathcal{S}}$ and $\boldsymbol{\mu}_{\boldsymbol{\theta}}^{\mathcal{S}}$, we decide whether to restart SVRE (by using the averaged iterate as the new starting point–Alg. 3, Line 6–$\bar{\boldsymbol{\omega}}_t$) or computing the batch snapshot at a point $\boldsymbol{\omega}_t$.

# F    Details on the implementation

For our experiments, we used the PyTorch[8] deep learning framework, whereas for computing the FID and IS metrics, we used the provided implementations in Tensorflow[9].

## F.1    Metrics

We provide more details about the metrics enumerated in § 4. Both FID and IS use: (i) the *Inception v3 network* (Szegedy et al., 2015) that has been trained on the ImageNet dataset consisting of $\sim 1$ million RGB images of 1000 classes, $C = 1000$. (ii) a sample of $m$ generated images $x \sim p_g$, where usually $m = 50000$.

**Algorithm 3** Pseudocode for Restarted SVRE.

---
1: **Input:** Stopping time $T$, learning rates $\eta_{\boldsymbol{\theta}}, \eta_{\boldsymbol{\varphi}}$, both players' losses $\mathcal{L}^G$ and $\mathcal{L}^D$, probability of restart $p$.
2: **Initialize:** $\boldsymbol{\varphi}, \boldsymbol{\theta}, t = 0$          ▷ $t$ is for the online average computation.
3: **for** $e = 0$ **to** $T-1$ **do**
4:     Draw $\texttt{restart} \sim \mathrm{B}(p)$.        ▷ Check if we restart the algorithm.
5:     **if** $\texttt{restart}$ **and** $e > 0$ **then**
6:        $\boldsymbol{\varphi} \leftarrow \bar{\boldsymbol{\varphi}}, \ \ \boldsymbol{\theta} \leftarrow \bar{\boldsymbol{\theta}}$ and $t = 1$
7:        $\boldsymbol{\varphi}^{\mathcal{S}} \leftarrow \boldsymbol{\varphi}$ and $\boldsymbol{\mu}_{\boldsymbol{\varphi}}^{\mathcal{S}} \leftarrow \frac{1}{|\mathcal{Z}|} \sum_{i=1}^{n} \nabla_{\boldsymbol{\varphi}} \mathcal{L}_i^D(\boldsymbol{\theta}^{\mathcal{S}}, \boldsymbol{\varphi}^{\mathcal{S}})$
8:        $\boldsymbol{\theta}^{\mathcal{S}} \leftarrow \boldsymbol{\theta}$ and $\boldsymbol{\mu}_{\boldsymbol{\theta}}^{\mathcal{S}} \leftarrow \frac{1}{|\boldsymbol{\varphi}|} \sum_{i=1}^{n} \nabla_{\boldsymbol{\theta}} \mathcal{L}_i^G(\boldsymbol{\theta}^{\mathcal{S}}, \boldsymbol{\varphi}^{\mathcal{S}})$
9:     $N \sim \mathrm{Geom}\left(1/n\right)$          ▷ Length of the epoch.
10:    **for** $i = 0$ **to** $N-1$ **do**
11:       **Sample** $i_{\boldsymbol{\theta}} \sim \pi_{\boldsymbol{\theta}}, i_{\boldsymbol{\varphi}} \sim \pi_{\boldsymbol{\varphi}}$, do **extrapolation:**
12:       $\tilde{\boldsymbol{\varphi}} \leftarrow \boldsymbol{\varphi} - \eta_{\boldsymbol{\theta}} \boldsymbol{d}_{\boldsymbol{\varphi}}(\boldsymbol{\theta}, \boldsymbol{\varphi}, \boldsymbol{\theta}^{\mathcal{S}}, \boldsymbol{\varphi}^{\mathcal{S}}) \ , \ \ \tilde{\boldsymbol{\theta}} \leftarrow \boldsymbol{\theta} - \eta_{\boldsymbol{\varphi}} \boldsymbol{d}_{\boldsymbol{\theta}}(\boldsymbol{\theta}, \boldsymbol{\varphi}, \boldsymbol{\theta}^{\mathcal{S}}, \boldsymbol{\varphi}^{\mathcal{S}})$    ▷ (63) and (64)
13:       **Sample** $i_{\boldsymbol{\theta}} \sim \pi_{\boldsymbol{\theta}}, i_{\boldsymbol{\varphi}} \sim \pi_{\boldsymbol{\varphi}}$, do **update:**
14:       $\boldsymbol{\varphi} \leftarrow \boldsymbol{\varphi} - \eta_{\boldsymbol{\theta}} \boldsymbol{d}_{\boldsymbol{\varphi}}(\tilde{\boldsymbol{\theta}}, \tilde{\boldsymbol{\varphi}}, \boldsymbol{\theta}^{\mathcal{S}}, \boldsymbol{\varphi}^{\mathcal{S}}) \ , \ \ \boldsymbol{\theta} \leftarrow \boldsymbol{\theta} - \eta_{\boldsymbol{\varphi}} \boldsymbol{d}_{\boldsymbol{\theta}}(\tilde{\boldsymbol{\theta}}, \tilde{\boldsymbol{\varphi}}, \boldsymbol{\theta}^{\mathcal{S}}, \boldsymbol{\varphi}^{\mathcal{S}})$    ▷ (63) and (64)
15:       $\bar{\boldsymbol{\theta}} \leftarrow \frac{t}{t+1} \bar{\boldsymbol{\theta}} + \frac{1}{t+1} \boldsymbol{\theta}$ and $\bar{\boldsymbol{\varphi}} \leftarrow \frac{t}{t+1} \bar{\boldsymbol{\varphi}} + \frac{1}{t+1} \boldsymbol{\varphi}$    ▷ Online computation of the average.
16:       $t \leftarrow t + 1$        ▷ Increment $t$ for the online average computation.
17: **Output:** $\boldsymbol{\theta}, \boldsymbol{\varphi}$

---

### F.1.1 Inception Score

Given an image $x$, IS uses the softmax output of the Inception network $p(y|x)$ which represents the probability that $x$ is of class $c_i, i \in 1 \dots C$, i.e., $p(y|x) \in [0,1]^C$. It then computes the marginal class distribution $p(y) = \int_x p(y|x) p_g(x)$. IS measures the Kullback–Leibler divergence $\mathbb{D}_{KL}$ between the predicted conditional label distribution $p(y|x)$ and the marginal class distribution $p(y)$. More precisely, it is computed as follows:

$$IS(G) = \exp\left(\mathbb{E}_{x \sim p_g}[\mathbb{D}_{KL}(p(y|x)||p(y))]\right) = \exp\left(\frac{1}{m} \sum_{i=1}^{m} \sum_{c=1}^{C} p(y_c|x_i) \log \frac{p(y_c|x_i)}{p(y_c)}\right). \quad (65)$$

It aims at estimating (i) if the samples look realistic i.e., $p(y|x)$ should have low entropy, and (ii) if the samples are diverse (from different ImageNet classes) i.e., $p(y)$ should have high entropy. As these are combined using the Kullback–Leibler divergence, the higher the score is, the better the performance. Note that the range of IS scores at convergence varies across datasets, as the Inception network is pretrained on the ImageNet classes. For example, we obtain low IS values on the SVHN dataset as a large fraction of classes are numbers, which typically do not appear in the ImageNet dataset. Since **MNIST** has greyscale images, we used a classifier trained on this dataset and used $m = 5000$. For the rest of the datasets, we used the original implementation[10] of IS in TensorFlow, and $m = 50000$.

### F.1.2 Fréchet Inception Distance

Contrary to IS, FID aims at comparing the synthetic samples $x \sim p_g$ with those of the training dataset $x \sim p_d$ in a feature space. The samples are embedded using the first several layers of the Inception network. Assuming $p_g$ and $p_d$ are multivariate normal distributions, it then estimates the means $\boldsymbol{m}_g$ and $\boldsymbol{m}_d$ and covariances $C_g$ and $C_d$, respectively for $p_g$ and $p_d$ in that feature space. Finally, FID is computed as:

$$\mathbb{D}_{\mathrm{FID}}(p_d, p_g) \approx d^2((\boldsymbol{m}_d, C_d), (\boldsymbol{m}_g, C_g)) = ||\boldsymbol{m}_d - \boldsymbol{m}_g||_2^2 + Tr(C_d + C_g - 2(C_d C_g)^{\frac{1}{2}}), \quad (66)$$

where $d^2$ denotes the Fréchet Distance. Note that as this metric is a distance, the lower it is, the better the performance. We used the original implementation of FID[11] in Tensorflow, along with the provided statistics of the datasets.

### F.1.3   Second Moment Estimate

To evaluate SVRE effectively, we used the **second moment estimate** (SME, uncentered variance, see § D.1) of the gradient estimate throughout the iterations $t = 1 \dots T$ per parameter, computed as: $v_t = \gamma v_{t-1} + (1-\gamma)g_t^2$, where $g_t$ denotes the gradient estimate for the parameter and iteration $t$, and $\gamma = 0.9$. For SVRE, $g_t$ is $d_{\boldsymbol{\varphi}}$ and $d_{\boldsymbol{\theta}}$ (see Eq. 63 and 64) for $G$ and $D$, respectively. We initialize $g_0 = 0$ and we use bias-corrected estimates: $\hat{v} = \frac{v_t}{1-\gamma^t}$. As the second moment estimate is computed per each parameter of the model, we depict the average of these values for the parameters of $G$ and $D$ separately.

In this work, as we aim at assessing if SVRE *effectively* reduces the variance of the gradient updates, we use SME in our analysis as it is computationally inexpensive and fast to compute.

### F.1.4   Entropy & Total Variation on MNIST

For the experiments on **MNIST** illustrated in Fig. 5a & 3b in § 4, we plot in § G the **entropy** (E) of the generated samples' class distribution, as well as the **total variation** (TV) between the class distribution of the generated samples and a uniform one (both computed using a pretrained network that classifies its 10 classes).

## F.2   Architectures & Hyperparameters

**Description of the architectures.**   We describe the models we used in the empirical evaluation of SVRE by listing the layers they consist of, as adopted in GAN works, e.g. (Miyato et al., 2018). With "conv." we denote a convolutional layer and "transposed conv" a transposed convolution layer (Radford et al., 2016). The models use Batch Normalization (Ioffe and Szegedy, 2015) and Spectral Normalization layers (Miyato et al., 2018).

### F.2.1   Architectures for experiments on MNIST

For experiments on the **MNIST** dataset, we used the DCGAN architectures (Radford et al., 2016), listed in Table 3, and the parameters of the models are initialized using PyTorch default initialization. We used mini-batch sizes of 50 samples, whereas for full dataset passes we used mini-batches of 500 samples as this reduces the wall-clock time for its computation. For experiments on this dataset, we used the *non saturating* GAN loss as proposed (Goodfellow et al., 2014):

$$\mathcal{L}_D = \mathbb{E}_{x \sim p_d} \log(D(x)) + \mathbb{E}_{z \sim p_z} \log(D(G(z))) \tag{67}$$
$$\mathcal{L}_G = \mathbb{E}_{z \sim p_z} \log(D(G(z))), \tag{68}$$

where $p_d$ and $p_z$ denote the data and the latent distributions (the latter to be predefined).

For both the baseline and the SVRE variants we tried the following step sizes $\eta = [1 \times 10^{-2}, 1 \times 10^{-3}, 1 \times 10^{-4}]$. We observe that SVRE can be used with larger step sizes. In Table 9, we used $\eta = 1 \times 10^{-4}$ and $\eta = 1 \times 10^{-2}$ for SE–A and SVRE(–VRAd), respectively.

### F.2.2   Choice of architectures on real-world datasets

We replicate the experimental setup described for **CIFAR-10** and **SVHN** in (Miyato et al., 2018), described also below in § F.2.4. We observe that this experimental setup is highly sensitive to the choice of the

| Generator |
|---|
| *Input:* $z \in \mathbb{R}^{128} \sim \mathcal{N}(0, I)$ |
| transposed conv. (ker: 3×3, 128 → 512; stride: 1) |
| Batch Normalization |
| ReLU |
| transposed conv. (ker: 4×4, 512 → 256, stride: 2) |
| Batch Normalization |
| ReLU |
| transposed conv. (ker: 4×4, 256 → 128, stride: 2) |
| Batch Normalization |
| ReLU |
| transposed conv. (ker: 4×4, 128 → 1, stride: 2, pad: 1) |
| $Tanh(\cdot)$ |

| Discriminator |
|---|
| *Input:* $x \in \mathbb{R}^{1 \times 28 \times 28}$ |
| conv. (ker: 4×4, 1 → 64; stride: 2; pad:1) |
| LeakyReLU (negative slope: 0.2) |
| conv. (ker: 4×4, 64 → 128; stride: 2; pad:1) |
| Batch Normalization |
| LeakyReLU (negative slope: 0.2) |
| conv. (ker: 4×4, 128 → 256; stride: 2; pad:1) |
| Batch Normalization |
| LeakyReLU (negative slope: 0.2) |
| conv. (ker: 3×3, 256 → 1; stride: 1) |
| $Sigmoid(\cdot)$ |

Table 3: DCGAN architectures (Radford et al., 2016) used for experiments on **MNIST**. We use *ker* and *pad* to denote *kernel* and *padding* for the (transposed) convolution layers, respectively. With $h \times w$ we denote the kernel size. With $c_{in} \to y_{out}$ we denote the number of channels of the input and output, for (transposed) convolution layers.

hyperparameters (see our results in § G.3), making it more difficult to compare the optimization methods for a fixed hyperparameter choice. In particular, apart from the different combinations of learning rates for $G$ and $D$, for the baseline this also included experimenting with: $\beta_1$ (see (56)), a multiplicative factor of exponential learning rate decay scheduling $\gamma$, as well as different ratio of updating $G$ and $D$ per iteration. These observations, combined with that we had limited computational resources, motivated us to use shallower architectures, which we describe below in § F.2.3, and which use an inductive bias of so-called Self–Attention layers (Zhang et al., 2018). As a reference, our SAGAN and ResNet architectures for **CIFAR-10** have approximately 35 and 85 layers, respectively–in total for G and D, including the non linearity and the normalization layers. For clarity, although the deeper and the shallower architectures differ as they are based on ResNet and SAGAN, we refer these as *deep* (see § F.2.3) and *shallow* (see § F.2.4), respectively.

### F.2.3 Shallower SAGAN architectures

We used the SAGAN architectures (Zhang et al., 2018), as the techniques of self-attention introduced in SAGAN were used to obtain the state-of-art GAN results on ImageNet (Brock et al., 2019). In summary, these architectures: (i) allow for attention-driven, long-range dependency modeling, (ii) use spectral normalization (Miyato et al., 2018) on both $G$ and $D$ (efficiently computed with the *power iteration* method); and (iii) use different learning rates for $G$ and $D$, as advocated in (Heusel et al., 2017). The foremost is obtained by combining weights, or alternatively *attention vectors*, with the convolutions across layers, so as to allow modeling textures that are consistent globally–for the generator, or enforcing geometric constraints on the global image structure–for the discriminator.

We used the architectures listed in Table 5 for **CIFAR-10** and **SVHN** datasets, and the architectures described in Table 6 for the experiments on **ImageNet**. The models' parameters are initialized using the default initialization of PyTorch.

For experiments with SAGAN, we used the hinge version of the adversarial non-saturating loss (Lim and Ye, 2017; Zhang et al., 2018):

$$\mathcal{L}_D = \mathbb{E}_{x \sim p_d} \max(0, 1 - D(x)) + \mathbb{E}_{z \sim p_z} \max(0, 1 + D(G(z))) \quad (69)$$
$$\mathcal{L}_G = -\mathbb{E}_{z \sim p_z} D(G(z))., \quad (70)$$

| **Self–Attention Block** ($d$ – input depth) | | |
|---|---|---|
| *Input:* $t \in \mathbb{R}^{d \times H \times W}$ | | |
| *i:* conv. (ker: $1{\times}1$, $d \to \lfloor d/8 \rfloor$) | *ii:* conv. (ker: $1{\times}1$, $d \to \lfloor d/8 \rfloor$) | *iii:* conv. (ker: $1{\times}1$, $d \to d$) |
| *iv:* softmax( *(i)* $\otimes$ *(ii)* ) | | |
| *Output:* $\gamma\big((iv) \otimes (iii)\big) + t$ | | |

Table 4: Layers of the self–attention block used in the SAGAN architectures (see Tables 5 and 6), where $\otimes$ denotes matrix multiplication and $\gamma$ is a scale parameter initialized with $0$. The columns emphasize that the execution is in parallel, more precisely, that the block input $t$ is input to the convolutional layers *(i)–(iii)*. The shown row ordering corresponds to consecutive layers' order, *e.g.* softmax is done on the product of the outputs of the *(i)* and *(ii)* convolutional layers. The $1 \times 1$ convolutional layers have stride of 1. For complete details see Zhang et al. (2018).

| **Generator** |
|---|
| *Input:* $z \in \mathbb{R}^{128} \sim \mathcal{N}(0, I)$ |
| transposed conv. (ker: $4{\times}4$, $128 \to 256$; stride: 1) |
| Spectral Normalization |
| Batch Normalization |
| ReLU |
| transposed conv. (ker: $4{\times}4$, $256 \to 128$, stride: 2, pad: 1) |
| Spectral Normalization |
| Batch Normalization |
| ReLU |
| Self–Attention Block (128) |
| transposed conv. (ker: $4{\times}4$, $128 \to 64$, stride: 2, pad: 1) |
| Spectral Normalization |
| Batch Normalization |
| ReLU |
| Self–Attention Block (64) |
| transposed conv. (ker: $4{\times}4$, $64 \to 3$, stride: 2, pad: 1) |
| $Tanh(\cdot)$ |

| **Discriminator** |
|---|
| *Input:* $x \in \mathbb{R}^{3 \times 32 \times 32}$ |
| conv. (ker: $4{\times}4$, $3 \to 64$; stride: 2; pad: 1) |
| Spectral Normalization |
| LeakyReLU (negative slope: 0.1) |
| conv. (ker: $4{\times}4$, $64 \to 128$; stride: 2; pad: 1) |
| Spectral Normalization |
| LeakyReLU (negative slope: 0.1) |
| conv. (ker: $4{\times}4$, $128 \to 256$; stride: 2; pad: 1) |
| Spectral Normalization |
| LeakyReLU (negative slope: 0.1) |
| Self–Attention Block (256) |
| conv. (ker: $4{\times}4$, $256 \to 1$; stride: 1) |

Table 5: *Shallow* SAGAN architectures for experiments on **SVHN** and **CIFAR-10**, for the Generator (left) and the Discriminator (right). The self-attention block is described in Table 4. We use the default PyTorch hyperparameters for the Batch Normalization layer.

where consistent with the notation above, $p_d$ and $p_z$ denote the data and the latent distributions.

For the SE–A baseline we obtained best performances when $\eta_G = 1 \times 10^{-4}$ and $\eta_D = 4 \times 10^{-4}$, for G and D, respectively. Similarly as noted for **MNIST**, using SVRE allows for using larger order of the step size on the rest of the datasets, whereas SE–A with increased step size ($\eta_G = 1 \times 10^{-3}$ and $\eta_D = 4 \times 10^{-3}$ failed to converge. In Table 2, $\eta_G = 1 \times 10^{-3}$, $\eta_D = 4 \times 10^{-3}$, and $\eta_G = 5 \times 10^{-3}$, $\eta_D = 8 \times 10^{-3}$, $\beta_1 = 0.3$ for SVRE and SVRE–VRAd, respectively. We did not use momentum for the vanilla SVRE experiments.

| Generator |
|---|
| *Input:* $z \in \mathbb{R}^{128} \sim \mathcal{N}(0, I)$ |
| transposed conv. (ker: $4{\times}4$, $128 \to 512$; stride: 1) |
| Spectral Normalization |
| Batch Normalization |
| ReLU |
| transposed conv. (ker: $4{\times}4$, $512 \to 256$, stride: 2, pad: 1) |
| Spectral Normalization |
| Batch Normalization |
| ReLU |
| transposed conv. (ker: $4{\times}4$, $256 \to 128$, stride: 2, pad: 1) |
| Spectral Normalization |
| Batch Normalization |
| ReLU |
| Self–Attention Block (128) |
| transposed conv. (ker: $4{\times}4$, $128 \to 64$, stride: 2, pad: 1) |
| Spectral Normalization |
| Batch Normalization |
| ReLU |
| Self–Attention Block (64) |
| transposed conv. (ker: $4{\times}4$, $64 \to 3$, stride: 2, pad: 1) |
| $Tanh(\cdot)$ |

| Discriminator |
|---|
| *Input:* $x \in \mathbb{R}^{3 \times 64 \times 64}$ |
| conv. (ker: $4{\times}4$, $3 \to 64$; stride: 2; pad: 1) |
| Spectral Normalization |
| LeakyReLU (negative slope: 0.1) |
| conv. (ker: $4{\times}4$, $64 \to 128$; stride: 2; pad: 1) |
| Spectral Normalization |
| LeakyReLU (negative slope: 0.1) |
| conv. (ker: $4{\times}4$, $128 \to 256$; stride: 2; pad: 1) |
| Spectral Normalization |
| LeakyReLU (negative slope: 0.1) |
| Self–Attention Block (256) |
| conv. (ker: $4{\times}4$, $256 \to 512$; stride: 2; pad: 1) |
| Spectral Normalization |
| LeakyReLU (negative slope: 0.1) |
| Self–Attention Block (512) |
| conv. (ker: $4{\times}4$, $512 \to 1$; stride: 1) |

Table 6: *Shallow* SAGAN architectures for experiments on **ImageNet**, for the Generator (left) and the Discriminator (right). The self–attention block is described in Table 4. Relative to the architectures used for **SVHN** and **CIFAR-10** (see Table 5), the generator has one additional "common" block (conv.–norm.–ReLU), whereas the discriminator has additional "common" block as well as self–attention block (both of more parameters).

### F.2.4 Deeper ResNet architectures

We experimented with ResNet (He et al., 2015) architectures on **CIFAR-10** and **SVHN**, using the architectures listed in Table 8, that replicate the setup described in (Miyato et al., 2018) on **CIFAR-10**. For experiments with ResNet, we used the hinge version of the adversarial non-saturating loss, Eq. 69 and 70. For this architectures, we refer the reader to § G.3 for details on the hyperparameters, where we list the hyperparameters along with the obtained results.

| G–ResBlock |
| --- |
| *Bypass*: |
| Upsample($\times 2$) |
| *Feedforward*: |
| Batch Normalization |
| ReLU |
| Upsample($\times 2$) |
| conv. (ker: $3\times3$, $256 \to 256$; stride: 1; pad: 1) |
| Batch Normalization |
| ReLU |
| conv. (ker: $3\times3$, $256 \to 256$; stride: 1; pad: 1) |

| D–ResBlock ($\ell$–th block) |
| --- |
| *Bypass*: |
| [AvgPool (ker:$2\times2$ )], if $\ell = 1$ |
| conv. (ker: $1\times1$, $3_{\ell=1}/128_{\ell\neq1} \to 128$; stride: 1) |
| Spectral Normalization |
| [AvgPool (ker:$2\times2$, stride:2)], if $\ell \neq 1$ |
| *Feedforward*: |
| [ ReLU ], if $\ell \neq 1$ |
| conv. (ker: $3\times3$, $3_{\ell=1}/128_{\ell\neq1} \to 128$; stride: 1; pad: 1) |
| Spectral Normalization |
| ReLU |
| conv. (ker: $3\times3$, $128 \to 128$; stride: 1; pad: 1) |
| Spectral Normalization |
| AvgPool (ker:$2\times2$ ) |

Table 7: ResNet blocks used for the ResNet architectures (see Table 8), for the Generator (left) and the Discriminator (right). Each ResNet block contains skip connection (bypass), and a sequence of convolutional layers, normalization, and the ReLU non–linearity. The skip connection of the ResNet blocks for the Generator (left) upsamples the input using a factor of 2 (we use the default PyTorch upsampling algorithm–nearest neighbor), whose output is then added to the one obtained from the ResNet block listed above. For clarity we list the layers sequentially, however, note that the bypass layers operate in parallel with the layers denoted as "feedforward" (He et al., 2015). The ResNet block for the Discriminator (right) differs if it is the first block in the network (following the input to the Discriminator), $\ell = 1$, or a subsequent one, $\ell > 1$, so as to avoid performing the ReLU non–linearity immediate on the input.

| Generator | Discriminator |
| --- | --- |
| *Input*: $z \in \mathbb{R}^{128} \sim \mathcal{N}(0, I)$ | *Input*: $x \in \mathbb{R}^{3\times32\times32}$ |
| Linear($128 \to 4096$) | D–ResBlock |
| G–ResBlock | D–ResBlock |
| G–ResBlock | D–ResBlock |
| G–ResBlock | D–ResBlock |
| Batch Normalization | ReLU |
| ReLU | AvgPool (ker:$8\times8$ ) |
| conv. (ker: $3\times3$, $256 \to 3$; stride: 1; pad:1) | Linear($128 \to 1$) |
| $Tanh(\cdot)$ | Spectral Normalization |

Table 8: *Deep* ResNet architectures used for experiments on **SVHN** and **CIFAR-10**, where G–ResBlock and D–ResBlock for the Generator (left) and the Discriminator (right), respectively, are described in Table 7. The models' parameters are initialized using the Xavier initialization (Glorot and Bengio, 2010).

|            | IS |  |  | FID |  |  |
|------------|-------|------|-----------|-------|-------|-----------|
|            | SE–A | SVRE | SVRE–VRAd | SE–A | SVRE | SVRE–VRAd |
| MNIST      | 8.62 | 8.58 | 8.56 | 0.17 | 0.15 | 0.18 |
| CIFAR-10   | 6.61 | 6.50 | **6.67** | 37.20 | 39.20 | 38.88 |
| SVHN       | 2.83 | 3.01 | **3.04** | 39.95 | 24.01 | **19.40** |
| ImageNet   | 7.22 | **8.08** | 7.50 | 89.40 | **75.60** | 81.24 |

Table 9: Best obtained IS and FID scores for the different optimization methods, using *shallow* architectures, for a fixed number of iterations (see § F). The architectures for each dataset are described in: **MNIST**–Table 3, **SVHN** and **CIFAR-10**–Table 5, and **ImageNet**–Table 6. The standard deviation of the Inception scores is around 0.1 and is omitted. Although the IS metric gives relatively close values on **SVHN** due to the dataset properties (see § F.1), we include it for completeness.

## G   Additional Experiments

### G.1   Results on MNIST

The results in Table 2 on **MNIST** are obtained using 5 runs with different seeds, and the shown performances are the averaged values. Each experiment was run for $100K$ iterations. The corresponding scores with the standard deviations are as follows: (i) IS: 8.62±.02, 8.58±.08, 8.56±.11;   (ii) FID: 0.17±.03, 0.15±.01, 0.18±.02; for SE–A, SVRE, and SVRE–VRAd, respectively.  On this dataset, we obtain similar final performances if run for many iterations, however SVRE converges faster (see Fig. 3). Fig. 5 illustrates additional metrics of the experiments shown in Fig. 3.

### G.2   Results with shallow architectures

Fig. 6 depicts the results on **ImageNet** using the *shallow* architectures described in Table 6, § F.2.3. Table 9 summarizes the results obtained on **SVHN**, **CIFAR-10** and **ImageNet** with these architectures. Fig. 7 depicts the SME metric (see § F.1.3) for the the SE–A baseline and SVRE shown in Fig. 3c, on **SVHN**.

(a) IS (higher is better)

(b) Entropy (higher is better)

(c) Total variation (lower is better)

(d) Discriminator

Figure 5: Stochastic, full-batch and variance reduced versions of the extragradient method ran on **MNIST**, see § 4.1. *BatchE–A* emphasizes that this method is **not** scaled with the number of passes (x-axis). The input space is $1 \times 28 \times 28$, see § F.2 for details on the implementation.

(a) IS (higher is better)

(b) FID (lower is better)

Figure 6: Comparison between *SVRE* and the *SE–A* baseline on **Imagenet**, using the *shallow* architectures described in Table 6. See § F.1 for details on the used IS and FID metrics.

(a) Generator

(b) Discriminator

Figure 7: Average second moment estimate (see § F.1.3) on **SVHN** for the Generator (left) and the Discriminator (right), using the *shallow* architectures described in Table 5. The corresponding FID scores for these experiments are shown in Fig. 3c.

### G.3 Results with deeper architectures

We observe that GAN training is more challenging when using *deeper* architectures and some empirical observations differ in the two settings. For example, our stochastic baseline is drastically more unstable and often does not start to converge, whereas SVRE is notably *stable*, but slower compared to when using shallower architectures. In this section, all our discussions focus on *deep* architectures (see § F.2.4).

**Stability: convergence of the GAN training.** For our stochastic baselines, *irrespective whether we use the extragradient or gradient method*, we observe that the convergence is notably more *unstable* (see Fig. 8) when using the *deep* architectures described in § F.2.4. More precisely, either the training fails to converge or it diverges at later iterations. When updating G and D equal number of times *i.e.* using $1:1$ update ratio, using SE–A on **CIFAR-10** we obtained best FID score of $24.91$ using $\eta_G = 2 \times 10^{-4}$, $\eta_D = 4 \times 10^{-4}$, $\beta_1 = 0$, while experimenting with several combinations of $\eta_G, \eta_D, \beta_1$. Using exponential learning rate decay with a multiplicative factor of $0.99$, improved the best FID score to $20.70$, obtained for the experiment with $\eta_G = 2 \times 10^{-4}$, $\eta_D = 2 \times 10^{-4}$, $\beta_1 = 0$. Finally, using $1:5$ update ratio, with $\eta_G = 2 \times 10^{-4}$, $\eta_D = 2 \times 10^{-4}$, $\beta_1 = 0$ provided best FID of $18.65$ for the baseline. Figures 8a and 8b depict the hyperparameter sensitivity of SE–A and SG–A, respectively. The latter denotes the alternating GAN training with Adam, that is most commonly used for GAN training.

(a) SE–A, **CIFAR-10**        (b) SG–A, **SVHN**

Figure 8: FID scores (lower is better) with different hyperparameters for the SE–A baseline on **CIFAR10** (left) and the SG–A baseline on **SVHN** (right), using the *deep* architectures described in Table 8, § F.2.4. SG–A denotes the standard stochastic *alternating* GAN training, with the Adam optimization method. Where omitted, $\beta_1 = 0$, see (56) where this hyperparameter is defined. With $r$ we denote the update ratio of generator versus discriminator: in particular $1:5$ denotes that $D$ is updated 5 times for each update of $G$. $\gamma$ denotes a multiplicative factor of exponential learning rate decay scheduling. In Fig. 8b, $\gamma = 0.99$ for all the experiments. We observed in all our experiments that training diverged in later iterations for the stochastic baseline, when using *deep* architectures.

We observe that SVRE is more stable in terms of hyperparameter selection, as it always starts to converge and *does not diverge* at later iterations. Relative to experiments with shallower architectures, we observe that with deeper architectures SVRE takes longer to converge than its baseline for this architecture. With constant step size of $\eta_G = 1 \times 10^{-3}$, $\eta_D = 4 \times 10^{-3}$ we obtain FID score of $23.56$ on **CIFAR-10**. Note that this result outperforms the baseline when using no additional tricks (which themselves require additional hyperparameter tuning). Fig. 9 depicts the FID scores obtained when training with SVRE on the **SVHN** dataset, for two

Figure 9: Obtained FID (lower is better) scores for SVRE, using the *deep* architectures (see § F.2.4) on **SVHN**. With $s$ we denote the fixed random seed. The update ratio for all the experiments is $1:1$. We illustrate our results on the same plot (besides the reduced clarity) so as to summarize our observation that, contrary to the SE–A baseline for these architectures, SVRE *always converges*, and does not diverge.

Figure 10: Obtained FID (lower is better) scores for WS–SVRE, using the *deep* architectures (see § F.2.4) on **CIFAR-10**, where the seed is fixed to $1$ for all the experiments. With $r$ we denote the update ratio of generator versus discriminator: in particular $1:5$ denotes that $D$ is updated $5$ times for each update of $G$. We start from the best obtained FID score for the stochastic baseline, i.e. FID of $18.65$ (see Table 2)–shown with dashed line, and we continue to train with SVRE.

different hyperparameter settings, using four different seeds for each. From this set of experiments, we observe that contrary to the baseline that either did not converge or diverged in all our experiments, SVRE always converges. However, we observe different performances for different seeds. This suggests that more exhaustive empirical hyperparameter search that aims to find an empirical setup that works *best* for SVRE or further combining SVRE with adaptive step size techniques are both promising research directions (see our discussion below). Fig. 10 depicts our WS–SVRE experiment, where we start from a stored checkpoint for which we obtained best FID score for the SE–A baseline, and we continue the training with SVRE. It is interesting that besides that the baseline diverged after the stored checkpoint, SVRE further reduced the FID score. Moreover, we observe that using different update ratios does not impact much the performance, what on the other hand was necessary to make the baseline algorithm converge.

(a) Generator                                                                (b) Discriminator

Figure 11: Average second moment estimate (SME, see § F.1.3) on **CIFAR-10** for the Generator (left) and the Discriminator (right), using the *deep* architectures described in Table 8. The obtained FID scores for these experiments are shown in Fig. 8a, where we omit some of the experiments for clarity. All of the baseline SE–A experiments diverge at some point, what correlates with the iterations at which large oscillations of SME appear for the Discriminator. Note that the SE–A experiments were stopped after the algorithm diverges, hence the plotted SME is up to a particular iteration for two of the experiments (shown in blue and orange). The SE–A experiment with $\gamma = 0.99$ diverged at later iteration relative to the experiments without learning rate decay, and has lower SME.

**Second moment estimate (SME).**  Fig. 11 depicts the second moment estimate (see § F.1.3) for the experiments with *deep* architectures. We observe that: (i) the estimated SME quantity is more bounded and changes more smoothly for SVRE (as we do not observe large oscillations of it as it is the case for SE–A); as well as that   (ii) divergence of the SE–A baseline *correlates* with large oscillations of SME, in this case, observed for the Discriminator. Regarding the latter, there exist larger in magnitude oscillations of SME (note that the exponential moving average hyperparameter for computing SME is $\gamma = 0.9$, see § F.1.3).

**Conclusion & future directions.**  In summary, we observe the following most important advantages of SVRE when using *deep* architectures: (i) consistency of convergence, and improved stability; as well as (ii) reduced number of hyperparameters. Apart from the practical benefit for applications, the former could allow for a more fair comparison of GAN variants. The latter refers to the fact that SVRE omits the tuning of the sensitive (for the stochastic baseline) $\beta_1$ hyperparameter (see (56)), as well as $r$ and $\gamma$–as training converges for SVRE without using different update ratio and step size schedule, respectively. It is important to note that the stochastic baseline does not converge when using constant step size (i.e. when *SGD* is used instead of *Adam*). In our experiments we compared SVRE that uses constant step size, with Adam, making

the comparison unfair toward SVRE. Hence, our results indicate that SVRE can be further combined with adaptive step size schemes, so as to obtain both stable GAN performances and fast convergence when using these architectures. Nonetheless, the fact that the baseline either does not start to converge or it diverges later makes SVRE and WS–SVRE a promising approach for practitioners using these *deep* architectures, whereas, for *shallower* ones, SVRE speeds up the convergence and often provides better *final* performances.