[Reviews · NeurIPS 2019]

Reviewer 1



This is an interesting and very well written paper. The observation that previous methods (such as extragradient) fail in the stochastic setting for minimax problems is novel and important,. The main algorithm (SVRE) aims to address this failure. My major comment is that the theoretical guarantees given in the paper need strong assumptions, e.g. the assumptions do not hold even for the counter-example provided in section 2.2. Can the authors find a counter-example that satisfies assumption 1 (i..e both L^G and L^D are strongly monotone). Also, can the authors provide some theoretical guarantee for the case where functions are non-convex?

Reviewer 2



Overall, I found this paper to be interesting and well-written. It was somewhat surprising to me that stochastic extragradient would not converge in cases where batch extragradient does, and the authors did a nice job of presenting the counterexample and explaining why this is the case. The proposed algorithm SVRE appears to work fairly well in practice, and the main theorem (Thm. 2) appears to be correct, and doesn't make excessively restrictive assumptions. I expect these results would be of interest to the community, in part because it combines two timely topics: variance reduction and optimization of games. One comment is that in the description of variance reduced gradient methods in page 4, I think you could have explained a little more about the work of Palaniappan and Bach and why their work is a natural model for your own. In particular, what are the main differences, and how does the analysis differ? I would have been curious to see a comparison of the IS (or something comparable) as a function of wall-clock time for BatchE in addition to SE. Is there any reason that was not included? In general, it seemed like the authors did not spend as much time/space evaluating the computational tradeoffs between the algorithms as a function of wall-clock time, which I think is ultimately the more important metric (as opposed to number of mini-batches, for instance). Another minor comment is that there are some strange grammatical errors in the introduction in particular, it may be worth reading over that to clean it up.

Reviewer 3



In this paper, authors first investigate the interplay between noise and multi-objective problems in the context, and then propose a new method ‚Äústochastic variance reduced extragradient‚ÄĚ (SVRE) which combines SVRG estimates of the gradient with the extragradient method (EG). This method succeeds to reduce noise in GAN training and improve upon the best convergence rates. advantages: Show that the noise can make stochastic extragradient with a motivating example. Combine the advantages of stochastic variance reduced extragradient method (SVRE) and extragradient method (EG). Experimentally, it effectively reduces the noise in GAN training. As shown in experiments (table2), it can improve SOTA deep models in the late stage of their optimization. disadvantages: In table2, SVRE has worst performance on CIFAR10. Authors do not give a explanation here but only show WS-SVRE, which apply SVRE from an iterate point of other method, has best performance. It will be better if the reason behind that is carefully explored.

[Author Response · NeurIPS 2019]

We thank the reviewers for their interest in the contributions of the paper and their detailed comments. We share
their enthusiasm regarding our theoretical contributions: we find fascinating how stochasticity can hurt convergence
in differentiable games and how variance reduction fixes it. We will revise our paper to reflect points raised in their
reviews (including the introduction, as noted by R3).

**R1 & R3: Extension of Theorem 1 that satisfies Assumption 1.** We obtain a result similar to Theorem 1 that satisfies
Assumption 1 by adding an $\ell_2$ penalty to Eq. 1, thus considering the following optimization problem:

$$\min_{\boldsymbol{\theta} \in \mathbb{R}^d} \max_{\boldsymbol{\varphi} \in \mathbb{R}^d} \frac{\epsilon}{2}\|\boldsymbol{\theta}\|^2 - \frac{\epsilon}{2}\|\boldsymbol{\varphi}\|^2 + \frac{1}{n}\sum_{i=1}^{n} \boldsymbol{\theta}^\top \boldsymbol{A}_i \boldsymbol{\varphi} \tag{1}$$

We can follow the same proof technique as in §C.1 and get a similar result as L88 with additional $\eta\epsilon$ terms:

$$\mathbb{E}[N_{t+1}] = \left(1 - \frac{|I|}{n}(2\eta\epsilon - \eta^2(1+\epsilon^2)) + \frac{|I|^2}{n^2}(2\eta^2(\eta\epsilon - 1) + \eta^4)\right)\mathbb{E}[N_t] \underset{|I|\ll n}{\approx} \left(1 - \eta\frac{|I|}{n}(2\epsilon - \eta(1+\epsilon^2))\right)\mathbb{E}[N_t].$$

Thus, for any step-size (roughly) larger than $2\epsilon$, the stochastic extragradient method diverges geometrically. However,
the full batch method [Harker and Pang] and SVRE (Thm. 2) do converge for any step-size smaller than 1 (particularly
for any step-size in $[2\epsilon, 1]$). This provides an example that satisfies Assumption 1 where the stochasticity breaks the
properties of extragradient (a step-size around $\epsilon$ would lead to a much slower convergence rate than for SVRE).

**R1: Guarantee for the non-convex case.** Recently, [Lin, Jin, and Jordan, 2019] provide guarantees in the min-max
setting when one of the two functions is non-convex and the other one is convex. Proving global convergence rate
when both $\mathcal{L}_G$ and $\mathcal{L}_D$ are non-convex in the full batch setting remains an open question that highly interests the
optimization community, but is outside the scope of this paper. As noticed by the reviewers, our goal was rather to
study the theoretical impact of stochasticity in convex games (and empirically for GANs).

**R3: Differences between this work and Palaniappan & Bach, and novelty.** We point out some of the differences in
lines 131–135, lines 164–174 and Table 1 of the paper. We agree that our algorithm may seem conceptually analogous
to the one of Palaniappan & Bach (which combines gradient method with variance reduction) as SVRE combines
extragradient with variance reduction. However, pointing out that stochasticity could be an important consideration
for solving the training instabilities of GANs is novel and there was no algorithm for extragradient that does variance
reduction. Also, it was not known whether extragradient would benefit (theoretically and practically) from variance
reduction. Our analysis largely differs from the one of P & B since the original analysis of extragradient is completely
different from the one of the gradient method. Precisely, the key point that allows for proving that the method has
a convergence rate of the order of $\mu/L$ (which is significantly better than the one in P & B) holds in Eq. 53 and 54.
Table 3 also compares SVRE with the existing standard methods. Regarding the practical contribution, we are excited
that SVRE resolves partially the known GAN training instabilities as, to our knowledge, SVRE is the first constant
step size method that works for non trivial datasets. Related works plug in *Adam* to make the algorithm work, which
unfortunately does not work consistently across hyperparameters for GANs and diverges at some point (see Fig. 8).

**R3: Wall-clock time for BatchE.** Fig. 1 below shows the wall clock time on **MNIST**, for a fixed GPU. The trend is
similar to Fig. 3a, where we used the number of mini-batches as a more portable comparison point (and standard in
optimization) enabling better reproducibility (since it is both hardware and implementation-independent) of the results.

**R4: More analysis and intuition in the experimental evaluation part.** Lines 252–257 & 294–301 discuss the main
points about the results from Tab. 2; while App. G.3 provides a more detailed discussion and additional experiments
(moved to App. due to space constraints). In short, SVRE might do worse than EG-A because the latter has the benefit
from adaptive step-sizes with Adam; developing a convergent adaptive step-size version of SVRE is an open problem.

Figure 1: Wall–clock time on **MNIST**, using **Tesla V100-SXM2-16GB** GPUs (see Appendix F.2.1 for experimental setup). We used $time.perf\_counter()$ & $torch.cuda.synchronize()$ to syncronize the cuda execution–following the recommendation for PyTorch, see the following link: *https://discuss.pytorch.org/t/best-way-to-measure-timing/39496*.

[Meta-Review · NeurIPS 2019]

The paper is certainly of interest to the NeurIPS community and very well written. However, the title is misleading and should be changed as for GAN the objective functions are not convex (or strongly convex). However, the result of this paper is only valid when such strong assumptions are made. Moreover, the counterexample provided by the authors does not even work under their own assumptions.